# Long-term trends of instability and associated parameters over the Indian region obtained using radiosonde network

Rohit Chakraborty, Madineni Venkat Ratnam* and Shaik Ghouse Basha

National Atmospheric Research Laboratory, India.

*Correspondence to:* M. Venkat Ratnam (vratnam@narl.gov.in)

**Abstract**

Long-term trends of the parameters related to convection and instability obtained from 27 radiosonde stations across 6 sub-divisions over the Indian region during the period 1980-2016 is presented. A total of 16 parcel and instability parameters along with moisture content, wind shear, and thunderstorm and rainfall frequencies have been utilized for this purpose. Robust fit regression analysis is employed on the regional average time series to calculate the long-term trends on both seasonal and yearly basis. The Level of Free Convection (LFC) and Equilibrium Level (EL) height is found to ascend significantly in all Indian sub-divisions. Consequently, the coastal regions (particularly the western coasts) experience increasing in Severe Thunderstorm (TSS) and Severe Rainfall Frequencies (SRF) in the pre-monsoon while the inland regions (especially central India) experience an increase in Ordinary Thunderstorm (TSO) and Weak Rain Frequency (WRF) during the monsoon and post-monsoon. The 16-20 year periodicity is found to dominate the long-term trends significantly compared to other periodicities and the increase in TSS, and Convective Available Potential Energy (CAPE) is found more severe after the year 1999. The enhancement in moisture transport and associated cooling at 100 hPa along with dispersion of boundary layer pollutants is found to be the main cause for the increase in CAPE which leads to more convective severity in the coastal regions. However, in inland regions moisture-laden winds are absent and the presence of strong capping effect of pollutants on instability in the lower troposphere has resulted in more Convective Inhibition Energy (CINE). Hence, TSO and weak rainfall occurrences have increased particularly in these regions.

*Key words*: Instability, Convection, Long-term trends, Radiosonde

## 1. Introduction

Intense convective phenomena are a common climatic feature in the Indian tropical region which occurs during the pre-monsoon to post-monsoon seasons (April–October) (Ananthakrishnan, 1977) and they are generally accompanied by intense thunderstorms, lightning, wind gusts with heavy rainfall. Hence, they are known to induce immense socio-economic hazards including loss of life and property. Several reports have shown an increase in the climatic extreme occurrence and intensity of these phenomena throughout the world (Webster et al., 2005; Emanuel, 2006). In this connection, the traditional surface-based parcel theory has been utilized to understand convective processes using atmospheric soundings as it calculates the atmospheric instabilities and other parameters at various heights (Huntrieser et al., 1997; Santhi et al., 2014; Nelli et al, 2018a).

Considering the importance of studying the long-term trends in climatic extremes, a series of research attempts have been orchestrated world-wide in the last two decades. Using multiple tropical stations and re-analysis data, Gettleman et al. (2002) and Riemann-Campe et al. (2009) have shown that Convective Available Potential Energy (CAPE) has been increasing very strongly with a growth rate of ~20% per decade during the period 1958-1997 due to increase in surface heating and moisture. Gensini and Mote (2015) projected a 236 % increase in severe thunderstorm frequency from 1980-1990 to 2080-2090 over the eastern United States (US). Further, Brooks (2013) used various combinations of CAPE and Vertical Wind Shear (VWSH) products and results hinted towards a probable increase in the severe thunderstorms over the US. It was also observed that the effect of increasing CAPE is more dominant on convective severity than in case of decreasing shear. On the other hand, studies by Prein et al. (2017) showed that a recent increase of temperature has led to a rise of moisture ingress and consequently the frequency and severity of extreme precipitation events associated with intense convection have shown a steep rise everywhere in the world. At the same time, an increase in thunderstorm severity and instability has also been reported by many attempts over the Asian region (Wang et al., 2011; Saha et al., 2017).

Over the Indian region, Manohar et al. (1999) studied the latitudinal variation and distribution of thunderstorm frequency and CAPE over 78 Indian stations during 1970-1980 and they postulated that the ambient temperature at 100 hPa pressure level has a strong relationship with it. Dhaka et al. (2010) utilized radiosonde observations during 1958-1997 and obtained very prominent anti-correlations on both yearly and seasonal basis between convection strength (CAPE) and upper troposphere temperatures at 100 hPa (T100). Later, Murugavel et al. (2012) studied the long term trends of CAPE from 32 radiosonde stations during 1984-2008 and revealed an alarming growth in monsoon CAPE over India with a slope of 38 J/kg/year. However, they additionally stated that the low-level moisture and solar cycle can have additional impact on the increasing CAPE. Recently from reanalysis datasets, Chakraborty et al. (2017a) and Saha et al. (2017) reported that lower lower tropospheric instability is reducing over few Indian stations after 1980 due to increasing levels of pollution. Apart from that, some studies have also attempted to correlate convective severity with boundary layer phenomena, surface fluxes, solar effect and precipitation; (Murthy and Sivaramakrishnan, 2006; Allappattu and Kunnikrishnan, 2009; Xie et al., 2011, Nelli et al. 2018b).

Previous studies over India have shown the distribution of CAPE only whereas other parameters like Convective Inhibition Energy (CINE), Mixed Layer CAPE (MLC), Lifted Index (LI), Total Totals Index (TTI), and Precipitable Water Vapor (PWV) are also important as they explain how the atmospheric instability and moisture changes at various levels of the atmosphere. In addition, the influence of climatic oscillation (Quasi-Biennial Oscillation (QBO), El-Nino Southern Oscillation (ENSO) and Solar Cycle) on the seasonal and annual variation of convective parameters was also not studied in detail. Therefore in the present study, long-term variation of parcel parameters (Lifted Condensation Level (LCL), Level of Free Convection (LFC), Equilibrium Level (EL), CAPE and CINE), with instability (LI, Vertical Totals Index (VT)), moisture (PWV, and PWV at low levels (PWL)), thunderstorm and rainfall severity frequencies (Thunderstorm-Severe (TSS), Thunderstorm-Ordinary (TSO), Weak Rain Fall (WRF) and Strong Rain Fall (SRF)) followed by Temperature at 100hPa (T100) and Wind Shear (WSH) is investigated using 27 radiosonde stations along with gridded rainfall data over India. This article is structured as follows: Section 2 describes the datasets and methodology adopted for the present study. Section 3 presents the long-term analysis of parcel and instability parameter over Chennai

(13.08°N, 80.27°E) and 6 sub-divisions of the Indian subcontinent on both annual and seasonal basis, followed
by the periodicity and split trend analysis. Finally, a discussion on the results and conclusions is appended in
Section 4 and 5, respectively.

**2. Dataset and Methodology**
Radiosonde observations from 27 stations over the Indian region from 1980-2016 are obtained from
Integrated Global Radiosonde Archives (https://www1.ncdc.noaa.gov/pub/data/igra/derived/derived-por/).
These datasets provide daily temperature and humidity profiles from 1538 stations around the world in fixed
pressure levels after doing quality checks (Durre et al., 2006; Ferreira et al. 2018). These studies have concluded
that the radiosonde data quality from IGRA has faced certain problems from time to time, but such cases are not
so prominent over the Indian region, especially after the year 1980. It is mainly because of the higher accuracy
and reliability of this in-situ measurement technique that these datasets are widely used worldwide nowadays for
calibrating other continuous profiler instruments (Chakraborty and Maitra, 2016). In accordance with data
availability and reliability, only 27 stations have been considered out of 37 IGRA Indian radiosonde stations
thereby providing descent data availability for carrying out this study. When an in depth investigation is done on
the data continuity by plotting the temperature and humidity profiles for all days, a set of gaps in datasets were
noticed. Most of the utilized stations have  intermittent data gaps of 2-7 days in certain months only but, on the
whole, except only a very few cases, the duration of  these individual data gaps are mostly limited to less than 1
month. However, these small data gaps are not expected to provide any significant impact on the long-term
seasonal or annual average variations of 37 years x 12 months span.
In addition to the data availability, homogeneity also acts as a common concern before using the data.
However, such issues should not be considered serious as all three types of homogeneity issues namely: volume,
instrument type and quality have been addressed before commencing the study. First, about 5000 radiosonde
profiles are available in majority of IGRA stations which are uniformly distributed among all years and seasons
(except monsoon); hence it provides a decent data volume for investigation of yearly trends. Secondly, the data
of all Indian IGRA stations come from a single type of IM-MK3 radiosondes which has not undergone any
change in radiosonde accuracies in the last years and so this addresses the instrument type related issue. Finally,
regarding data quality, a set of 7 quality checks are performed by IGRA before accepting the data which should
remove any unreliable observations before being used in the study. These 7 quality checks also include
repetition check which rejects any possible case of humidity sensor saturation errors during rainy conditions
especially in monsoon. Thus it can be inferred that the obtained climatic trends of instability from IGRA is
expected to be far more reliable compared to other data sources.
These 27 stations have been divided into six homogenous regions as defined by the India
Meteorological Department (IMD) (Rao, 1976) which are: Central India (CI), East Coast (EC), North East (NE),
North West (NW), Peninsula India (PI) and the West Coasts (WC) as shown in Fig. 1.  Further, for simplicity,
these regions have again been combined into three major categories namely: coastal regions (EC and WC),
Inland (CI and PI) and others (NE and NW). After retrieving the profiles some more additional internal quality
checks are performed before using the data for every station. First, the balloon burst height has to be minimum
of 15 km to be selected for analysis. Second, any gaps of temperature and humidity in important pressure levels
such as 850, 700, 500, 300, 200 and 100 hPa will produce difficulty in calculation of atmospheric instability;
hence those profiles were rejected in quality control tests for all stations. Again, out of the available radiosonde
profiles, some profiles have displayed absurd variations of temperature and humidity at various heights and
hence they are discarded. After completion of these quality checks it was thought that atmospheric instability
shows prominent diurnal variation, datasets of only one time slot can be taken for analysis. Finally as datasets at
00Z are consistently much more in number than at 12Z, hence analysis has been actually done with 00Z
datasets. A complete detail about the final dataset used for every station is indicated in Table A1. It may be
noted here that the volume of observations are found to distributed almost homogenously throughout the
measurement period and a detailed year wise breakup of radiosonde launches utilized are not shown to maintain
the focus of the work. For calculation of the instability parameters, the temperature and humidity profiles were
transformed from the standard pressure levels using cubic spline interpolation at every 100 m height bins. Piece-
wise linear/ quadratic/ cubic spline interpolation schemes are employed instead of linear interpolation in
temperature and humidity retrievals in this study as the former techniques can more faithfully regenerate the
nonlinearities in boundary layer variations of meteorological parameters according to recent studies by
Chakraborty et al. (2016).. After this, a similar surface-based parcel method is utilized for estimating the parcel
and instability parameters (LCL, LFC, EL, CAPE, MLC, CINE) as already described by Chakraborty et al.
(2018). A small detail about the physical significance of these parameters is now given in the Supplementary
Section. For thunderstorm genesis, moisture growth and wind shear are extremely important, therefore we
calculated the total amount of water vapor (PWV) and that up to 700 hPa level (PWL) along with the horizontal
wind shear between surface and 6 km altitude. In addition to these, we have used temperature at the 100 hPa
pressure level as it is found to strongly influence the convective strengths over the Indian region (Manohar et al.,
1999; Dhaka et al., 2010).
Along with these parameters, the long-term impact of instability on the convection has also been
studied from thunderstorm and rain frequencies. Daily measurements of surface wind speeds is obtained for all
the radiosonde observations at 00Z using the Wyoming Website (weather.uwyo.edu/upperair/sounding.html).
The thunderstorm frequencies are calculated on yearly basis based on the criterion given by IMD
(http://imd.gov.in/section/nhac/termglossary.pdf). According to this criterion, if the maximum surfaces wind
speed is greater than 62 km/h then it is considered as a severe thunderstorm event otherwise if wind speeds are
between 31 and 62 km/h then it is considered as an ordinary thunderstorm case (also used by Saha et al. (2014)).
Hereafter, the total number of thunderstorm occurrences per year in both severe and ordinary category is
counted and represented by thunderstorm frequencies as TSS and TSO. Here it may be noted that, the wind
speed measurements are taken from the first measurement of radiosonde balloon flight for all stations. These
datasets are always within 10m from the surface and according to WMO criterion, they can assume a maximum
error of 1 m/s from surface to 100 hPa level. Since a minimum wind speed of 31kmph or 8.61 m/s is required
for identification as an ordinary thunderstorm, hence this 1 m/s error is not expected to perturb the thunderstorm
severity climatology presented in this study.
IMD provides daily rainfall accumulations in 0.25 degree spatial resolution over the Indian region since
the year 1900 (Rajeevan et al., 2006, 2008; Pai et al., 2014). This daily precipitation data at the closest grid point
is used to define the frequency of severe and weak rainfall days hereafter referred to as WRF and SRF
respectively. The severe rainfall frequencies constitute those days where the daily accumulation is greater than
124.5 mm/day while for the weak rainfall cases it is less than 7.5 mm/day according to IMD glossary as given in
http://imd.gov.in/section/nhac/termglossary.pdf .

From the previous section it follows that a set of 14 parcel parameters with rainfall and thunderstorm

frequencies are essential to understand the convective climatology over India. However, other than this, 8
standard instability parameters (LI, KI, TTI, CT, VT, CAPE, CINE and MLC) are also additionally important to
quantify the thunderstorm severity, hence must also be considered for analysis. Now, it is known that most of
these instability parameters are inter-related; hence principal component analysis (PCA) analysis is done to
identify and use only those instability parameters that can give a complete but independent overview of the
atmospheric instability using minimum parameters. In this analysis, introduced by (Hoteling 1936) a set of
possibly related parameters are converted into orthogonal independent components after which the primary
components are plotted with the initial parameters. Parameter variance scores present at the farthest distance
from the primary principal components and also from all the other variables contain the highest variance; hence
they are selected for representing the existing group of old inputs. Hence in the present study, daily datasets of
all 6 instability parameters are averaged to yearly values for every regions and then the PCA analysis is
performed on the datasets. Daily datasets have not been directly used for PCA as it would have too many
fluctuations which would make the redundant parameter identification very difficult in all cases. The variance
distribution plot (not shown) for each region showed that only the first two components contribute to more than
70% of the total variance; hence the covariance scores of these two strongest orthogonal components are plotted
in Fig. 2 which depict that the LI is completely unrelated to rest of the parameters. Again, since VT is found to
lie exactly in the middle of the rest of the parameters, and it also represents the lower tropospheric instability in
a much more suitable way hence this parameter is also used with LI to represent the rest of the instability
parameters in a convenient way.. Consequently, LI and VT are additionally considered along with the previous
set of 14 attributes to get the final set of 16 parameters for further analysis.

Thus, a set of 16 parameters are finally taken for the analysis: LCL, LFC, EL, LI, VT, CAPE, CINE,

MLC, PWV, PWL, WSH, T100, TSO, TSS, WRF and SRF. However, apart from the IGRA radiosonde and the
IMD rainfall database, it was believed that some other parameters may also be externally responsible for the
changing trends in atmospheric instability and hence they are also included. They comprise the monthly mean
aerosol absorption index (AAI) data taken from the Tropospheric Emission Monitoring Internet Service
(TEMIS) Air Pollution Archive (De Graaf et al., 2005). In addition, the monthly average gridded data of ozone
mixing ratio (OMR) and Specific Humidity (SHUM) along with Downward Long Wave Radiation Flux
(DLWRF)      are      also      utilized      from      ERA-Interim      Re-analysis      datasets
(https://apps.ecmwf.int/datasets/data/interim-full-daily/levtype=sfc/).

We have estimated all parameters from daily radiosonde data and averaged over a season and annually

for obtaining trend at 95% confidence interval using robust regression analysis (Shepard, 1968). Further, the
parameters from radiosonde were averaged region wise and then the robust fit algorithm is employed on the
normalized time series to get the long-term trends (Andersen, 2008; Raj et al., 2018). These yearly trend values
are multiplied by 37 to get the total climatological trend in one parameter over the complete data span of 1980-
2016. For seasonal trend analysis, the same approach has been utilized for different seasons. The seasonal
distribution has been adopted from IMD reports which are as follows: Pre-monsoon (March-May), Monsoon
(June-September), Post-monsoon (October-November) and Winter (December-February). Further, for studying
the periodicities associated with each of these time series, an Empirical Mode Decomposition (EMD) technique
is used (Wu and Huang, 2009). Finally, the robust fit analysis is done on each of these components to compare
the trends from each periodicity to determine which of the periodicities dominates in each parameter.
**3. Results**
**3.1. Climatic trends over Chennai**
In the previous study by Chakraborty et al. (2018), long term trends of instability were investigated over
Gadanki (13.5$^o$N,79.2$^o$E) situated on a hilly terrain with an altitude of 370 m above sea level at a distance of
~150 km from the eastern coasts and Bay of Bengal. To see whether, the observed trends of these parameters are
behaving similarly in case of IGRA profiles also the climatic trends of instability are now described over
Chennai (13.08$^o$N, 80.27$^o$E) which is the closest radiosonde station from Gadanki. The yearly averaged datasets
are normalized with respect to their climatic mean and are plotted along with 1 sigma standard errors in Fig 3
after which robust fit regression analysis (Andersen 2008) is utilized to obtain the climatological trends in these
parameters as shown by red solid lines in the plots. A decreasing trend in VT and increase of magnitude in
CINE with LI is noticed which indicates a reduction in the lower atmospheric instability (Fig.3d,e,h,i).
However, CAPE (Fig.3f) shows significant increasing trends throughout the period. LFC has a slightly
ascending trend (~18 hPa) which leads to increasing CINE and decreasing VT over Chennai, while the EL is
found to get lifted up drastically (Fig.3c) resulting in an increase in the total instability and CAPE. The increase
in height of EL can be caused by a reduction in temperatures in the upper tropospheric heights (Manohar et al,
1999). Hence, it can be inferred that the reduction in temperatures near 100 hPa (Fig.3l) plays an important role
in modulating the total atmospheric instability and CAPE.
The enhancement in CINE magnitude and reduction in VT leads to the reduction in the frequency of
weaker convective systems with medium or lower CAPE values. Again, as CAPE is one of most important
parameters that modulate convective severity, hence the frequency of severe thunderstorms and heavy rainfall
occurrences is expected to rise (Fig.3n,p). Thus, it is inferred that lower level instability has reduced due to
elevated CINE and LFC; while the upper-level atmospheric instability has intensified significantly due to a
cooling at 100 hPa and ascension in EL over Chennai. Hence, CAPE value increases drastically leading to more
severe thunderstorm and heavy rainfall frequency events during the mentioned period.
Before proceeding to the investigation on the climatological trends of convection and instability over
the Indian region, it is necessary to validate whether the obtained hypothetical trends from Chennai are free
from any data quality issues. Hence a region wise climatology of the most important parameter CAPE is
obtained from all the Indian regions using ERA-Interim Reanalysis data and the trends are shown in Fig. S1.
This figure clarifies that all the Indian regions (especially the coastal regions) have experienced a common rise
in CAPE especially after 1996-2000. Thus, the stated hypothesis looks clear and hence this can be progressed
over a much broader way.
However, it should be noted that Fig 3 provides too much detailed and cumbersome results related to
all 16 parameters and the complexity of the analysis is expected to increase further when similar analysis will be
presented for all the Indian regions together. On the other hand, for a complete understanding about the
morphology of upper and lower tropospheric instability, all the instability parameters will be required. Hence, to
reduce chances of confusion and to make the results more compact, all 16 parameters will be discussed together
but only a few of them will be presented in the main study. After a thorough consideration with respect to the
main objective of the present attempt, 8 parameters namely LFC, EL, CAPE, CINE, PWV, T100, TSS and SRF
are retained in the main figures while their complementary aspects such as LCL, LI, VT, MLC, PWL, WSH,
TSO and WRF are shown in the supplementary sections.
**3.2. Climatological average of parameters**
The climatological mean values of all instability parameters over six different Indian sub-divisions are
shown with boxplot analysis (McGill, 1978) in Fig. 4 and Fig. S3. The utility of using this approach is that, it
will reveal which regions of India shows normal expected variation (if it lies within the box limits signifying 25-
75% percentage of the distribution), while on the other hand it will also identify those regions having the
extreme outlier values (lying outside the whiskers  signifying the outermost 5% of the distribution). The LCL
(Fig.S3a) and LFC (Fig.4a) are found to be at the lowest altitudes in the coastal regions. As these stations
receive most of the moisture from Sea, the EL (Fig.4b) is also expected to be higher at the coastal areas and
lower elsewhere. However, due to low moisture availability, the inland regions experience weaker instability
which results in lower CAPE (~900 J/kg) (Fig.4c) with higher CINE (Fig.4d) and WSH (Fig.S3f). During strong
convection, the values of LI (Fig.S3b) (which represents that the mid-tropospheric instability) are also expected
to be more negative in the coastal regions. Similarly, height integrals of instability such as CAPE (Fig.4c) and
MLC (Fig.S3d) are significantly higher (~1500 J/kg) in the coastal regions while the magnitude of MLC
(Fig.S3d) is found to be almost half of CAPE. As the trends in total convective strengths below 300 hPa are
quite low compared to that over the total atmospheric column, hence it follows that the portion of buoyant
column above 300 hPa must have contributed significantly to the total convective developments over the Indian
region. Again, being opposite of CAPE, CINE values are minimum in the coastal regions compared to inland
and continental regions thereby serving as a potential cause for the reduced instability in these regions.
Similar to CAPE and MLC combination, the PWV (Fig.4e) and PWL (Fig.S3e) pair shows the highest
averages in the coastal regions due to their closest proximity to the adjoining seas. Also, PWL (moisture integral
up to 700 hPa) is found to be almost half of PWV, hence the mid and upper tropospheric humidity is found to
play a strong role in modulating the convective systems over India. The instability and moisture are highest in
the coastal regions hence the frequency of severe thunderstorms and rainfall occurrences are comparatively
higher (Fig.4g,h). The North Western region shows the large values of thunderstorm frequency which is not
supported by other parameters. Hence, it may be inferred that this is due to frequent dry storms called "Andhi"
which have no relation with convective instability and rainfall (Rajpal and Deka, 1980). Thus, it can be
concluded that the effect of convection is large in the coastal regions compared to other regions which resulted
in high CAPE with more thunderstorms and intense rain occurrences.
**3.3. Long-term trends in the instability parameters**
The long-term trends are calculated for each parameter during the entire study period of 1980-2016 for all
regions using the robust regression analysis at 95% confidence interval as depicted in Fig. 5 and Fig. S4. For
simplicity, the average trends along with their standard deviation values are depicted in Table 1. Also to
investigate about the significance of trend values calculated from these time series datasets, a *t-tset* analysis
(Gosset, 1908) is done on all parameter and locations. The p values are calculated at 95% confidence limits for
t-test analysis on all instability parameters over the Indian sub-divisions and interestingly, all the values are
found to be below 0.05. Hence the time series variations to be presented in subsequent sections will always be
statistically significant in nature. So, to have a better quantitative measure of the trend significance, the total
changes in each of these parameters are presented in percentage form in place of the p values in the table. This
process will enable an easy identification of regions experiencing more accelerated convective growth. But on
the other hand, while analyzing the results of the trend analysis in statistical form, the absolute trend has to be
given more importance as the % changes completely depend on the magnitude of the long term mean. The LCL
(Fig.S4a) height is found to decrease which may lead to an overall increase in the number of rain occurrences
throughout the country (provided that the amount of atmospheric instability is adequate). On contrary, LFC
(Fig.5a) is found to ascend in all the regions except NE resulting in the reduction of lower level instability and
an increase of CINE magnitude (Fig.5d). However, the extent of change in LFC (Fig.5a) and LCL (Fig.S4a) is
smallest in the coastal regions (~10 hPa). In case of EL (Fig.5b), a very prominent ascent is depicted in all
regions (highest in coastal regions) which increase the height of the buoyant column; hence the net effect on
total instability and CAPE (Fig.5c) is expected to increase significantly. Similarly, LI (Fig.S4b) values become
more negative in all the regions with slightly higher magnitudes in the coastal regions. VT represents the lower
level atmospheric instability and hence is expected to be affected by the elevation in LFC. Thus, a reduction in
VT (Fig.S4c) is seen with minimum values in the coastal regions (~0.3), medium in the NE and NW regions
(~0.5) and highest in deep inland regions such as CI and PI (~0.8). An intensification in CAPE (Fig.5c) is seen
in all regions (~1100 J/kg) as expected from EL (Fig.5b) and LI. However, the increase is the highest (~100%)
at the coastal regions whereas in MLC (Fig.S4d), which is measured only up to 300 hPa level, the increment is
only 20% of that in CAPE. Hence, it follows that the maximum contribution towards the increase in CAPE
comes above 300 hPa. In case of CINE, an overall enhancement in values is observed as expected (~60 J/kg). In
addition, the trend values suggest a two-fold increase of CINE in inland regions while the values are much lesser
(50%) in the coastal regions due to balancing effect from strong convections and CAPE in those regions.

The PWV (Fig.5e) and PWL (Fig.S4e) values are increasing similar to CAPE and MLC. The long-term
trends in PWV are about 10% of its climatological average with highest in the coastal regions. Further, the
lower level moisture content of PWL (up to 700 hPa) showed an increase but the trend values are comparatively
smaller (~6%). As it has been made clear that it is not the lower tropospheric moisture (below 700 hPa) but the
remaining amount which is increasing significantly at par with CAPE for all regions, hence there may be a
possible association between these two factors which needs to be investigated in the coming sections. The WSH
(Fig.S4f) parameter increases in all regions of the country, and hence it produces an inhibiting effect on the
lower level instability. An upper tropospheric cooling trend is observed in all other regions (Fig. 5f) with
minimum values in the inland regions and maximum in the coastal regions. Consequently, the increase in CAPE
values is maximum in the coastal regions and lesser elsewhere. The ordinary thunderstorm frequency is also
found to increase (Fig. S4g) which may be due to the partial damping effect of an elevated LFC and CINE on
lower level instabilities. However, the TSS (Fig.5g) is found to increase at a much higher rate compared to TSO
especially in the coastal regions. On the other hand, an increase in CINE and decrease of VT lead to an increase
in the number of WRF (Fig.S4h). However, due to rise in CAPE and TSS (Fig.5g), the SRF (Fig.5h) is also
found to rise significantly by about 20% particularly in the coastal regions. It may be noted that, as EL has more
dominant effect on CAPE hence the rise in SRF is much larger than that WRF (5%). Finally, the long-term
trends have been compared between the east and west coastal regions and it is observed that the rate of increase
in total instability is the most prominent in the western coasts while factors related to ascending LFC, CINE and

reducing VT are more significant in the central India which is the farthest from both the sea coasts. Thus, the long-term analysis infers that lower atmospheric instability has reduced while the upper tropospheric instability and moisture increased drastically over the Indian region. As a result, convective severity as expressed in terms of CAPE, TSS and SRF is increasing more strongly in the coastal regions while in the continental areas this effect is dampened due to the contribution of increasing CINE and WSH.

**3.4. Seasonal effect on long-term trends in the instability parameters**

The seasonal variation of the long-term variations of atmospheric instability is shown in Fig. 6 and Fig. S5. LCL shows a uniform descent by 10 hPa in all seasons (Fig.S5a) whereas LFC ascends in most of the regions and seasons (Fig.6a). However, this ascent is more prominent in the monsoon and post-monsoon season. However, the seasonal variation is absent in EL and LI (Fig.6b, Fig S5b) which are mainly associated with an upper layer phenomenon. VT shows the most prominent reduction in monsoon and post-monsoon seasons (Fig.S5c). MLC and CAPE show a lot of regional disparities but with a common increase in its value in all the seasons (Fig.6c, S5d). In monsoon and post-monsoon, the increase in CAPE is slightly lesser due to the effect of decreasing VT and elevated LFC. CINE is closely related to VT and LFC, hence it shows slight increase (of magnitude) in the monsoon and post-monsoon seasons with maximum values in inland regions as expected (Fig.6d).

PWV, PWL and WSH represent a prominent increase in the monsoon followed by the post-monsoon (Fig.6e, Fig.S5e-f). T100 is related to an upper atmospheric phenomenon hence no seasonal or spatial variation is displayed, except for a small cooling effect in pre-monsoon (Fig.6f) due to the prevalence of intense convections events which is supported by the strongest increase in CAPE. A decrease in lower atmospheric instability and increase in CINE is observed; hence TSO and WRF are expected to increase. However, this increase is found more dominant only in the monsoon and post-monsoon (Fig.S5g,h). Another interesting result is that TSS and SRF do not behave similarly. TSS increases almost uniformly in all seasons with the highest in the pre-monsoon. However, SRF increases mainly in the monsoon followed by the post-monsoon season (Fig.6g,h). The observed disparity between them is due to the profuse moisture availability during monsoon and post-monsoon compared to the pre-monsoon.

Further, in seasonal trends, east and west coasts show equivalent trends in all instability parameters while the Central India still remains as the region which is most affected by the ascension of LFC and CINE. Thus, the seasonal analysis reveals that the yearly long-term trends are almost uniformly distributed in all the seasons. The ordinary and weak thunderstorm frequencies show the strongest increase during monsoon and post-monsoon while the upper atmospheric instability shows a weak influence in the pre-monsoonal trends on the yearly climatology.

**3.5. Effect of specific periodicities on long-term trends**

In case of both annual and seasonal trend analysis, all Indian sub-divisions are found to follow similar behavior. Hence, to find out the periodicities in the average long-term trends, the time series of all regions are averaged and then subjected to EMD technique which reveals the existence of four main periodicities namely: 1.5 - 2.5 years corresponding to QBO, 4-6 years corresponding to ENSO, 10-12 years corresponding to the solar cycle and the fourth one is of 16-20 years. A similar multi-decadal climatic oscillation was also reported by Dhaka et al. (2010). Hence for simplicity, this periodicity has been renamed as a Multi-decadal Climatic Oscillation (MCO).

The climatic trends of these periodicities for each parameter are calculated from robust regression analysis. An illustration of the obtained MCO periodicities for CAPE along all the Indian regions is shown in Fig. S2. Further for comparison, the trend values from each periodicity is normalized to percentage with respect to the total trend values for each parameters and the net contribution of these individual periodicities are depicted in Fig. 7 and Fig. S6. The figure suggests that ENSO, QBO and solar cycle have no effect on LCL (Fig.S6a) while MCO is quite strong. LFC shows minimal effects to all periodicities except solar cycle period which may be due to solar-terrestrial heating (Fig.7a). EL and LI are significantly affected by both solar and MCO periodicities (Fig.7b, Fig, S6b). But in LI, the contribution from MCO is much more than solar effect. In case of VT (Fig. S6c) the effect of both ENSO and MCO are found prominent. CAPE is found to be strongly influenced by MCO followed by solar effect (Fig.7c) and this is also discernible from the most strong increasing trends in CAPE especially in the coastal regions after the years 1996-2000 in Fig. S2. However, in case of MLC, contribution of MCO is comparatively lesser (Fig.S6d) hence some separate phenomena above 300 hPa may have prominent influence on increasing CAPE. Apart from CAPE, effect of MCO is also found very strong in case of CINE (Fig.7d).

The moisture parameters like PWV and PWL show similar variability as in CAPE and MLC which indicates significant moisture transport changes only above 300 hPa in the past 18 years (Fig.7e, Fig S6e). WSH does not show the dominance of any periodicity (Fig.S6f) while T100 shows the most prominent contribution from the MCO (Fig.7f) thereby showing its connection with the long-term variability in EL and CAPE with associated thunderstorm severity in the recent years. TSO and TSS are both affected by solar and MCO (Fig.S6g, Fig.7g) but TSS shows that the effect of MCO is higher compared to TSO. Finally, the effect of MCO is also found more prominent in case of SRF and WRF (Fig.7h, Fig S6h). In nutshell, the MCO acts as the most dominant periodicity which has influenced the convective severity over India. So, in the coming sections, the MCO trends for both halves of 37 years will be studied, For ease of indication and referencing, these trends of 18 years span each will be hereafter mentioned as quasi-bi-decadal trends (since both spans are close to 20 years in length).

**3.6. Investigation of quasi-bi-decadal trends between 1980-1997 and 1999-2016**

In the previous section, the annual averaged time series of many parameters such as EL, LI, VT, CAPE, CINE, T100, TSS, WRF and SRF showed very significant changes with respect to MCO. It has also been indicated from Fig. S2 that the climatic trends before and after the period 1996-2000 are significantly different from each other. Therefore, the trends have been estimated with respect to two time periods before and after the year 1998. The time series for both MCO are produced and their trend values are represented in Fig. 8 and Fig. S7. For simplicity, the MCO are referred as C1 (1980 to 1998) and C2 (1999 to 2016), respectively. Starting with LCL, in C1 there is almost no change, but in C2 there is a strong descent which influences the overall change in the time series (Fig.S7a). In case of LFC, C1 shows an ascending trend, but in C2, a significant increasing pattern of LFC pressure is seen hence an overall descent is obtained (Fig.8a). An ascent in the EL is noticed in both the periods however during C2 the trends show significant enhancement (Fig.8b). LI values are expected to become more negative from 37 years trend, however its absolute magnitude shows a slight reduction in C1 followed by a prominent increase in C2 resulting in a net increase in instability (Fig.S7b). VT shows an overall decreasing pattern in both the periods (Fig.S7c). CAPE (Fig.8c) shows an enhancement in both the

cycles but the trends become more prominent in C2 (1500 J/Kg). Similar to CAPE, MLC (Fig.S7d) also shows an increasing trend in both the cycles but the trend values are also much smaller than in CAPE. Hence, the rise in EL height can be considered as a primary factor for increase in CAPE above 300 hPa during C2. CINE shows increasing trend in both C1 and C2 but again the trend values are much stronger (~80 J/kg) during C2 especially in the inland regions (Fig.8d)..

The moisture trends in both PWV and PWL have shown a constant increase in both the MCO throughout India (Fig.8e, Fig S7e). The WSH (Fig.8k) also increases uniformly in both MCO with strongest trends in the inland regions. A prominent cooling of ~1.5 degrees is seen in 100 hPa levels everywhere in C1, but in C2 the trend increases to ~-2.5 degrees (Fig.8f) which can be considered responsible for the abrupt elevation in EL and increasing CAPE values during the recent years. TSO increases slightly in C2 compared to C1 (Fig.S7g). But in case of TSS, the positive trend gets doubled in C2 mainly in the coastal regions (Fig.8g). Finally, in case of SRF the trend values in C2 are slightly higher with the maximum magnitudes in the coastal regions as expected (Fig.8h). A further comparison between the six regions reveals that the west coast shows the maximum enhancement in all the instability and convective severity parameters in the past 18 years due to strong growth in moisture content and associated cooling at 100 hPa.

On the contrary, during C2 central India suffers from the maximum reduction in lower level instability as seen from the rise in CINE and LFC due to the dearth of moisture content. Similar results are also found in other coastal and inland regions. Hence it follows that mainly during C2, the upper tropospheric instability has enhanced everywhere while the lower tropospheric instability has reduced which has led to the development in both CAPE and CINE. As a result both TSS-TSO and WRF -SRF combination increases.

## 4. Discussion

From the previous section, it is inferred that a cooling trend at 100 hPa levels has led to the ascent in EL which results in an increase in CAPE, TSS and SRF. To explain the reason behind this, we consider the ozone to be a primary heating agent by absorbing the incoming solar ultraviolet radiation near 100 hPa level (Mohanakumar, 2008). OH hydroxyl radicals are formed by oxidation of water vapor molecules by a reactive oxygen atom at the same height. On the other hand, it has been reported by Forster et al. (2007) that in the recent years there has been a cooling in upper troposphere due to decrease in ozone concentration near 70 hPa. Hence, it is hypothesized that the OH radicals formed from the oxidation of water vapour can take an active role in the breakup of ozone molecules at 100 hPa levels which may lead to this cooling effect. The preceding sections have shown a significant increase in moisture content especially in the coastal areas hinting towards more moisture transport from the adjoining seas. Again, an increase in LI and CAPE values have also been reported in most of the regions which can lift the available moisture to upper atmospheric levels (Das et al., 2016; Guha et al., 2017). To add to this increasing CAPE and LI, many recent researchers' have reported a net increase in the Hadley cell and Brewer-Dobson circulation strength (Liu et al., 2012; Fu et al., 2015; Shepherd and McLandress, 2011) which also assists in the up-liftment of moisture to upper atmospheric levels. Thus, it is inferred that low-level moisture is transported to the upper troposphere and above where it is responsible for ozone depletion and cooling thereby elevating the EL and increasing the thunderstorm severity.

To test this hypothesis, yearly averaged time series of specific humidity and ozone mixing ratio data are collected for all stations and the quasi-bi-decadal trend values are depicted in Fig. 9. This figure shows a rise in

specific humidity levels by 7% in C2 over entire India (Fig.9a). On the other hand, trends of specific humidity have almost trebled in C2 phase with the maximum values in the coastal regions (Fig.9e). As water vapor concentration increases, ozone concentration is expected to decrease. The ozone trends support this hypothesis by showing a sharp transition from low positive to high negative values during C2 (Fig.9b,f). It may be additionally noted that the specific humidity increase and reduction in ozone content are strongest in the coastal regions leading to a higher increase in CAPE and severe thunderstorms in those regions.

In the recent decades, Indian region has experienced a surface warming trend which is mainly caused by an increase in greenhouse gas concentrations as pointed out by Basha et al. (2017). These greenhouse gases are heat absorbing in nature and these particles reside within the lower troposphere (generally below 700 hPa) due to surface heating and boundary layer dynamics as reported by Chakraborty et al. (2017b). Further, these gases has a tendency to absorb and then re-emit the outgoing longwave radiation as emitted by the Earth resulting in more downward longwave radiation flux and atmospheric heating which elevates the LFC. Additionally, this near surface heating reduces the vertical temperature lapse rate leading to a drop in lower instability (VT). To test this hypothesis, yearly averaged Downward Long Wave Radiation Flux (DLWRF) time series is depicted over the Indian region in Fig. 9(c,g) which also suggests that DLWRF values are increasing in C2. To show that the increase in DLWRF is due to the heat absorbing particles only, the trends in Absorbing Aerosol Index (AAI) are shown for all the regions. The figure suggests that the mean of AAI is increasing slightly more in C2 with a positive trend (Fig.9d,h). Due to this heating of lower atmosphere and capping of lapse rates by greenhouse gases and absorptive aerosols, the LFC starts ascending, so WSH and CINE get stronger while VT reduces. As a result the ordinary to weak convective occurrences start increasing.

Finally, it has to be explained why the upper air instability and CAPE are increasing mainly in the coastal regions. The coastal regions have high moisture content (Saha et al., 2017)**.** Because of the strong land-ocean contrast, low-level winds close to 850 hPa flow into the coastal regions and disperse the pollutants and greenhouse gases to other locations leading to a weaker convective inhibition in those areas. This hypothesis is supported by the lowest AAI values in the coasts despite having high increasing trends in those areas. In addition, the ample moisture supply in the coastal regions is lifted up to the upper troposphere and lower stratosphere (UTLS) where it undergoes prominent cooling due to ozone reduction. Hence, the EL ascends more resulting in higher CAPE which finally led to an abrupt rise in TSS and SRF in the coastal regions. However, in the inland regions the layer of absorptive aerosols and greenhouse gases cannot be dispersed amply due to the dearth of strong lower level winds. As a result, the growth of lower atmospheric instability gets inhibited in the inland regions. Further, due to less moisture availability, UTLS cooling and EL ascent are much lower hence there is a less rise in CAPE which ultimately leads to an increase in TSO and WRF in those sub-divisions. It may be noted that the trend in AAI is not significantly different for the two time periods C1 and C2. Again, it is the EL and not the LFC or LCL which influences CAPE strongly; hence the strong trends of humidity increase and ozone reduction overpowers the weaker inhibitory effect from the atmospheric aerosols and this acts as a major driving force behind the increase in convective severity compared to in most of the cases.

**5. Summary and conclusions**

In recent decades, global warming has become a threat to human life and society in terms of its various implications. Increase in surface temperature leads to stronger atmospheric instabilities which in turn may

increase the CAPE resulting in more severe thunderstorm and precipitation activity. Hence, the long term
variations of instability parameters will help to better understand the changes in the weather extremes with
respect to climate change. In light of the above, the main objective of the present study is to analyze whether
convective instability is changing over the Indian region during the last 37 years, and then to find its possible
effects on thunderstorm and rainfall severity. Radiosonde measurements from Integrated Global Radiosonde
Archives (IGRA) pertaining to 27 stations across 6 Indian sub-divisions are utilized to depict the spatial
distribution of these long-term trends during the period of 1980-2016. The selection of instability parameters is
done based on Principle Component Test (PCT) which showed the importance of taking LI and VT for further
investigations. A total of 16 parameters (including parcel and instability data with moisture content, wind shear,
and thunderstorm and rainfall frequencies) have been utilized. Robust fit approach is employed on the regional
average time series to calculate the long-term trends on both yearly and seasonal basis. The main highlights
obtained from the present study are listed below:
1. The coastal regions experience the most significant rise in Convective Available Potential Energy (CAPE)
and Equilibrium Level (EL) leading to more occurrences of Severe Thunderstorms (TSS) and severe
rainfall events (SRF) while the inland regions undergo a decrease in lower atmospheric instability due to
elevated Convective Inhibition Energy (CINE) and Level of Free Convection resulting in more Ordinary
thunderstorm (TSO) and Weak Rainfall occurrences (WRF).
2. In the pre-monsoon season, an increasing TSS activity is observed due to higher instability connected to
increasing EL height and CAPE values, along with a decrease in LI values while, the monsoon and
postmonsoon season experiences more prominent ascension in LFC height with larger values of CINE,
Wind Shear (WSH) thereby increasing the tropospheric stability which lead to increased TSO and WRF
occurrences all over the Indian region.
3. The Empirical Mode Decomposition (EMD) analysis on the instability parameters reveals that the 16-20
year multi-decadal oscillation (MCO) as the most dominant component in all six Indian sub-divisions.
4. The quasi-bi-decadal analysis reveals an increase in magnitude in many parameters like EL, CAPE, CINE,
TSO and TSS along with cooling at 100 hPa level during C2 (1999-2016) which dominates 37-year trend.
5. The annual and quasi-bi-decadal trends support that the increase in thunderstorm severity and associated
convection is strongest along western coasts due to maximum moisture ingress from the seas while the
greatest reduction in lower atmospheric instability is experienced in central India owing to the lack of
pollutant dispersal as it is situated very far from the seas.
6. In the coastal regions, ample amount of water vapor is advected into the mid-troposphere from the
surrounding seas which in presence of strong lifting goes up to upper troposphere and lower stratosphere
(UTLS) where ozone depletion occurs leading to a strong cooling effect. This cooling effect enables the
ascent in EL resulting in much stronger LI and CAPE values, hence more TSS and SRF..
7. In the inland regions, the dispersing effect by sea winds is absent hence the capping effect of lower
instability is more leading to stronger CINE values. Again, due to the dearth of moisture transport from the
seas, the UTLS cooling is lesser; leading to a weaker rise in CAPE consequently the TSO and WRF
frequencies increase significantly.
8. However, as the ascent in EL has a stronger contribution over increasing CAPE than the inhibitory effect
of LFC, hence the long term trends are expected to be more strongly influenced by the ozone
decomposition and cooling at 100 hPa levels than the capping effect of low level inversions from
absorptive aerosols; hence the convective severity over the Indian regions is found to increase.
Thus, it may be inferred that in the near future also, convective severity will increase strongly in the
coastal regions while weak and ordinary thunderstorms will be more common in the inland regions. It may
appear at certain sections of this analysis that the trends of CAPE and EL are exorbitantly high; but it is not the
actual case because previous studies by Murugavel et al (2012) and Gettlemann et al. (2002) have also shown
almost comparable trends in convective severity both in India and abroad. Nevertheless, this study gets an upper
hand over the previous approaches as it successfully explains the hypothesis brought forward by early research
attempts that UTLS cooling at 100 hPa and greenhouse gases concentration rise can regulate the climatic trends
of convective severity and frequency especially over tropical regions in the recent decades.
After going through the study, there may be a possibility of thinking that the change in instability
trends is due to the change in sensors around 1998. But this is not the actual case because first, there has not
been any mention in past literature survey related to any change in radiosonde data quality during late 1990s in
IMD or IGRA. Secondly, the yearly variations of all 16 parameters for various IGRA stations as in Chennai do
not commonly show any abrupt change in time series during 1996-2004 except for a few cases. Thirdly it has
been revealed by IMD reports that the year 2000 was a tipping point for the climate change led warming over
India thereby leading to a rise in catastrophic weather events and a cataclysmic fallout will follow by the year
2040 if these emission scenarios are not curbed recently (Hindustan Times, 2019). Thus, it follows that the
observed changes in the atmospheric instability trends before and after 1996-2004 are due to a synoptic global
warming based climate change phenomena and not due to any change in radiosonde sensor type.
However, in spite of all this, the present study has certain shortcomings. The most important one
among them is that this set of explanations is based on isolated information from selected in-situ observations
and hence it needs to be studied in more detail spatially in future using model-based observations. In the recent
years, certain studies have utilized multiple GCM outputs over the US to infer the robust increase in
thunderstorm frequency (Diffenbaugh et al., 2013; Seeley and Romps, 2015). However, these types of studies
have not yet been done over the Indian region. Hence, a combination of multi-station radiosonde data with
model data will be utilized to provide a generalized picture about convective severity over the Indian region. On
the other hand, this study also introduces the effect of direct aerosol heating on instability and convection; but
the probable impact of indirect aerosol loading on modulating the cloud lifetime and convective severity has not
been discussed here. This is because, the relationship between indirect aerosol forcing and instability is still very
unclear and complex (Connoly et al. 2012). A few researches in the recent years have hypothesized that a higher
concentration of aerosols may lead to stronger updrafts velocities by altering the latent heat release resulting in
growth of CAPE and TSS (Tao et al. 2012; Storer and van den Heever, 2013). However, this is a season and
location specific phenomena and hence it is not expected to impact the yearly trend of CAPE and TSS as strong
as the upper tropospheric cooling effect projected in this study. But in future, an exhaustive analysis of cloud
and aerosol components involving both in-situ and modelled data can to be done to investigate its contribution
on the total CAPE, TSS and SRF trends over the Indian region.

**Author contributions:** Rohit Chakraborty had analysed complete data and written first draft, Ghouse Basha helped in the analysis and corrections and Venkat Ratnam supervised overall the work including final corrections.

**Acknowledgments**

One of the authors (Rohit Chakraborty) thanks, Science and Engineering Research Board, Department of Science and Technology for providing fellowship under National Post-Doctoral Scheme (File No:PDF/2016/001939). He also acknowledges National Atmospheric Research Laboratory, for providing necessary support and data for this work. The authors also thank S.T. Akhil Raj, Sanjeev Dwivedi and N. Narendra Reddy for their suggestions. Data used in present study can be obtained directly from IGRA website.

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

| Name | | CI | | | EC | | | NE | | | NW | | | PI | | | WC | | | India | | |
|---|---|---|---|---|---|---|---|---|---|---|---|---|---|---|---|---|---|---|---|---|---|---|
| | | μ | σ | % | μ | σ | % | μ | σ | % | μ | σ | % | μ | σ | % | μ | σ | % | μ | σ | % |
| LCL | (hPa) | 39 | 2.8 | 4.43 | 8.9 | 0.1 | 0.93 | 15.4 | 0 | 1.71 | 24 | 0.6 | 2.63 | 45.5 | 1.0 | 5.17 | 9.1 | 0.2 | 0.96 | 23 | 6.4 | 2.49 |
| LFC | (hPa) | -38 | 1.4 | 5.57 | -11 | 0.2 | 1.41 | 13.4 | 0.1 | 1.91 | -44 | 3.8 | 6.63 | -9.2 | 0.0 | 1.36 | -17 | 0.6 | 2.19 | -18 | 8.7 | 2.46 |
| EL | (hPa) | -188 | 11.2 | 49.2 | -280 | 27.1 | 82.8 | -206 | 8.8 | 60.5 | -230 | 2.0 | 67.6 | -239 | 6 | 63.6 | -311 | 40 | 91.4 | -242 | 19 | 68.1 |
| LI | ($^o$C) | -0.8 | 0.01 | 16.7 | -1.7 | 0.2 | 22.2 | -1.3 | 0.1 | 22.8 | -1.1 | 0.1 | 17.7 | -1.4 | 0.0 | 27.2 | -1.8 | 0.1 | 24.6 | -1.3 | 0.1 | 20.3 |
| VT | ($^o$C) | -0.7 | 0.01 | 2.98 | -0.3 | 0.02 | 1.50 | -0.5 | 0 | 2.24 | -0.5 | 0.0 | 2.32 | -0.9 | 0.1 | 3.85 | -0.4 | 0.0 | 1.95 | -0.5 | 0.1 | 2.34 |
| CAPE | (J/kg) | 617 | 2.9 | 82.8 | 1589 | 90.8 | 108 | 1137 | 30 | 125 | 858 | 53 | 90.3 | 1000 | 39 | 107 | 1554 | 198 | 98.9 | 1126 | 159 | 97.9 |
| MLC | (J/kg) | 55 | 0.24 | 12.1 | 288 | 9.6 | 42.8 | 273 | 4.8 | 56.7 | 134 | 1.1 | 29.7 | 201 | 16 | 43.7 | 323 | 27 | 42.0 | 212 | 42 | 36.8 |
| CINE | (J/kg) | -94 | 7.4 | 87.8 | -36 | 0.3 | 46.7 | -30 | 1.2 | 27.8 | -85 | 6.9 | 103 | -67 | 2.4 | 62.6 | -44 | 1.5 | 55.3 | -59 | 11 | 73.7 |
| PWV | (mm) | 1.4 | 0.03 | 5.71 | 3.2 | 0.03 | 10.0 | 3.7 | 0.1 | 13.4 | 1.3 | 0.0 | 4.72 | 2.2 | 0.0 | 8.97 | 3.9 | 0.1 | 11.2 | 2.6 | 0.5 | 8.85 |
| PWL | (mm) | -0.2 | 0 | 1.95 | 0.4 | 0 | 2.75 | 0.7 | 0.1 | 6.22 | 0.0 | 0 | 4.65 | 0.6 | 0.0 | 5.71 | 0.6 | 0.0 | 3.98 | 0.4 | 0.2 | 3.32 |
| WSH | (/s) | 5.8 | 0.3 | 78.4 | 3.4 | 0.2 | 54.4 | 4.8 | 0.2 | 75.0 | 3.4 | 0.0 | 54.4 | 5.5 | 0.6 | 74.8 | 3.6 | 0.1 | 68.6 | 4.4 | 0.4 | 69.8 |
| T100 | ($^o$C) | -1.5 | 0.03 | 3.00 | -2.5 | 0.1 | 5.20 | -0.4 | 0 | 0.83 | -0.3 | 0 | 0.59 | -2.5 | 0.3 | 5.13 | -2.2 | 0.2 | 4.68 | -1.6 | 0.4 | 3.00 |
| TSO | | 1.4 | 0.05 | 24.3 | 0.8 | 0 | 10.5 | 2.2 | 0.0 | 53.1 | 3.5 | 0.2 | 53.8 | 2.7 | 0.2 | 40.9 | 0.5 | 0 | 4.92 | 1.8 | 0.5 | 27.3 |
| TSS | | 1.5 | 0.0 | 70.5 | 2.3 | 0.05 | 144 | 2 | 0.1 | 250 | 2.5 | 0.3 | 209 | 1.7 | 0.1 | 81.7 | 2.3 | 0.1 | 131 | 2.1 | 0.2 | 147 |
| WRF | | 2.9 | 0.1 | 9.51 | 3.8 | 0.1 | 6.55 | 4.6 | 0.4 | 11.5 | 2.2 | 0.0 | 4.19 | 2.8 | 0.1 | 8.88 | 6.3 | 0.2 | 11.4 | 3.8 | 0.6 | 7.28 |
| SRF | | 0.4 | 0.0 | 32.4 | 0.8 | 0.06 | 22.2 | 0.2 | 0.0 | 8.30 | 0.2 | 0 | 14.8 | 0.2 | 0 | 14.4 | 1.1 | 0.1 | 39.5 | 0.5 | 0.2 | 20.5 |


**Table 1: Statistical information related to the 37-year trend of all instability parameters over the six sub-divisions of India** (μ: long-term average, σ: standard deviation, %: total percentage trend).

**Figures**

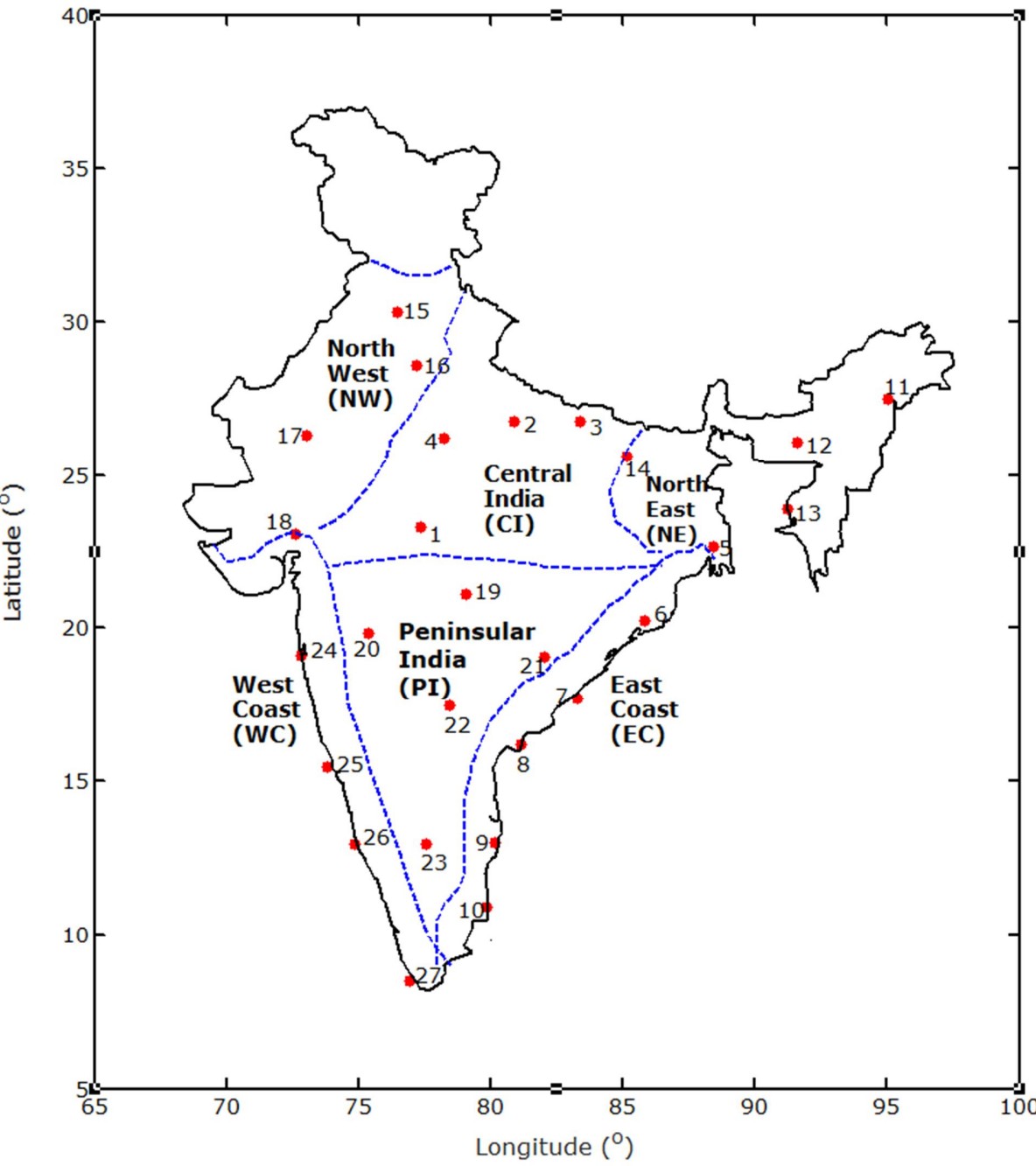

**Figure 1:** The locations of the 27 IGRA stations used for the present study. The distribution of the 27 stations over Indian regions is as follows: 4 stations in the NC, 6 stations in EC, 4 stations in NE, 4 stations in the NW, 5 stations in the PI and finally 4 stations in WC.

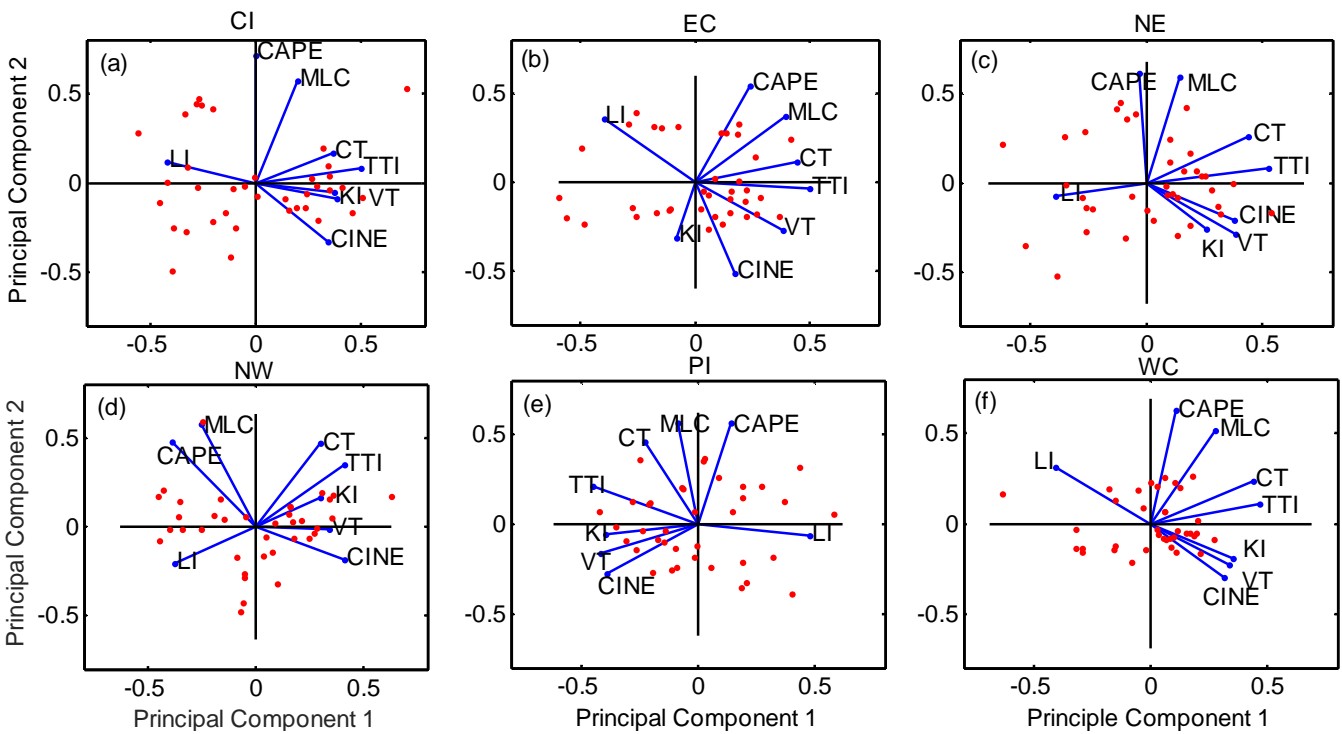

**Figure 2:** Principle Component Analysis for selection of instability parameters for the long-term trend study in (a) Central India (CI), (b) East Coast (EC), (c), North East (NE), (d) North West (NW), (e) Peninsular India (PI) and (f) West Coasts (WC) obtained using IGRA observations.

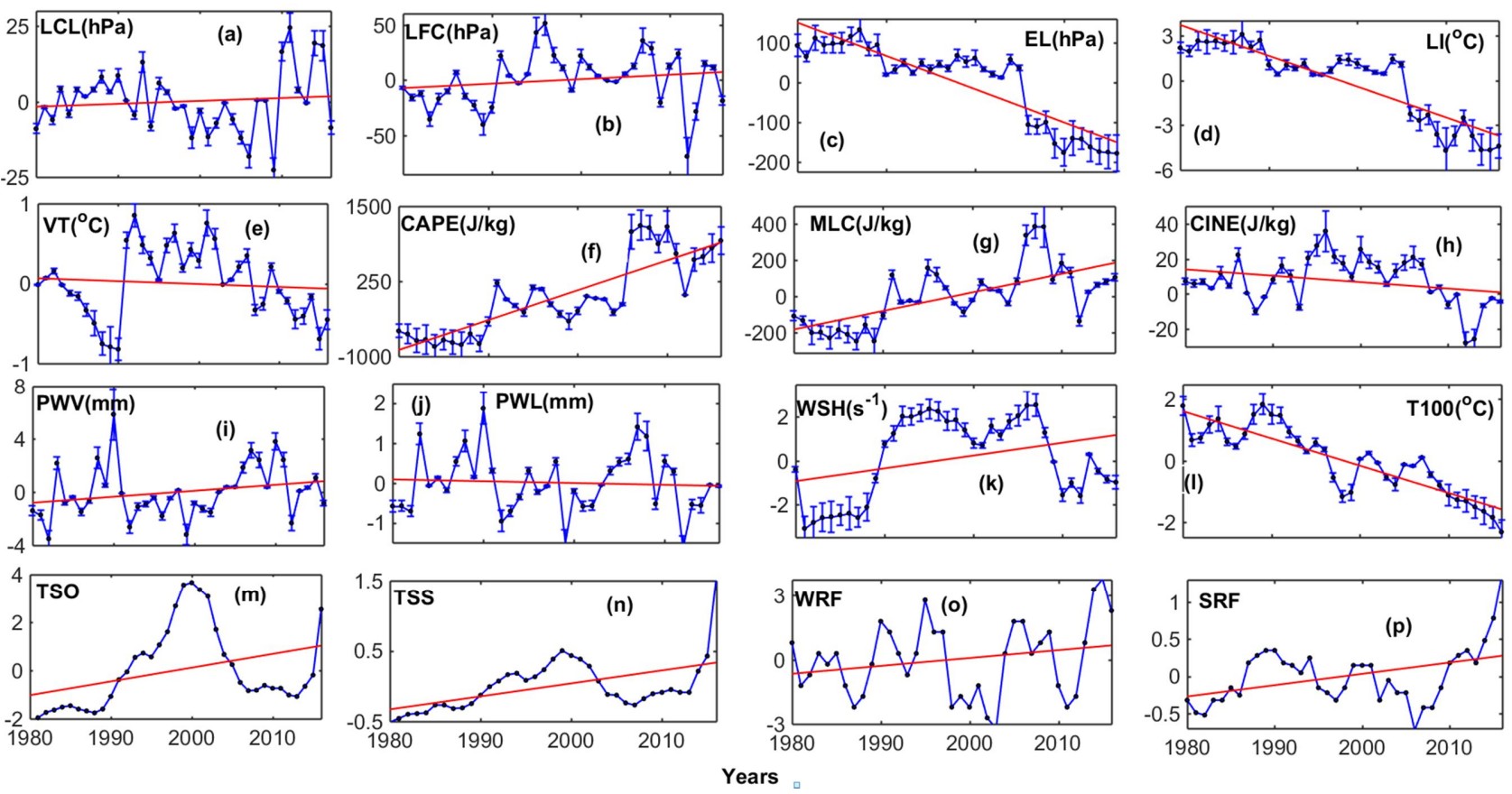

**Figure 3:** Long-term variation of (a) LCL, (b) LFC, (c) EL, (d) LI, (e) VT, (f) CAPE, (g) MLC, (h) CINE, (i) PWV, (j) PWL, (k) WSH, (l) T100, (m) TSO, (n) TSS, (o) WRF and (p) SRF observed over Chennai.

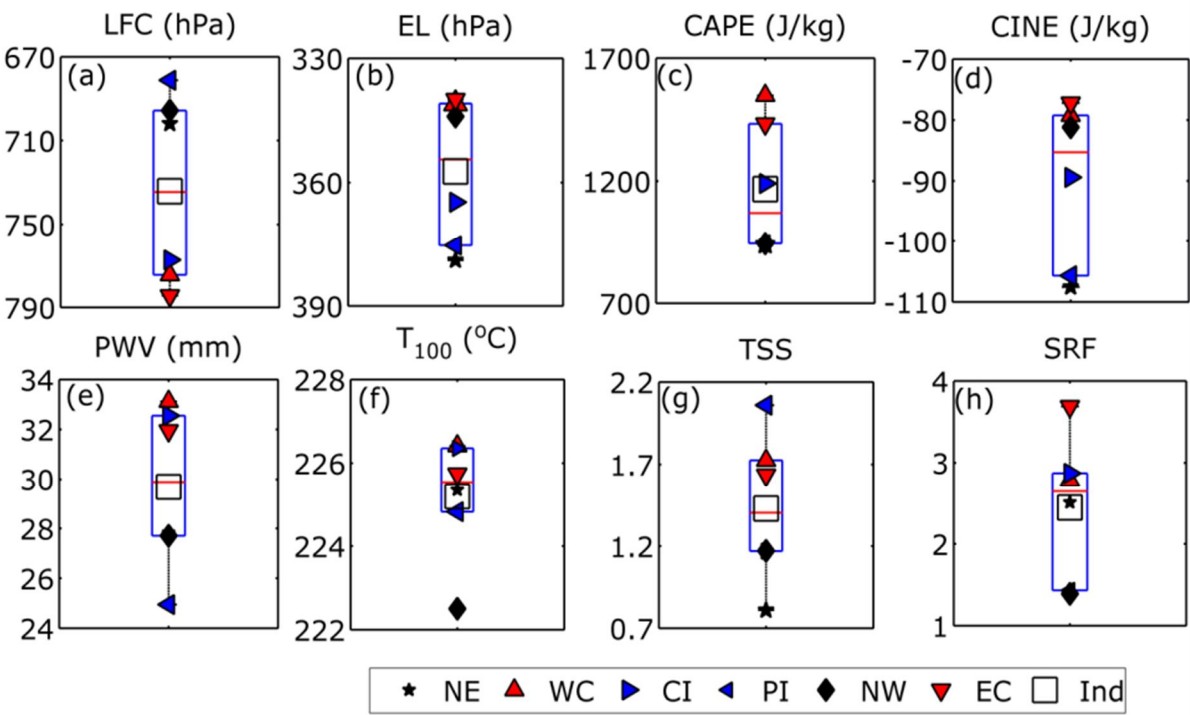

**Figure 4:** Climatological mean values of (a) LFC, (b) EL, (c) CAPE, (d) CINE, (e) PWV, (f) T100, (g) TSS and (h) SRF over the six sub-divisions of India. Coastal Regions are represented by red cones, the north eastern and western regions are denoted by black stars and diamonds while the blue cones represent the inland regions. Here the box limits refer to the upper and lower quartiles (25% and 75%) while the whiskers refer to the outlier limit of the data (5% and 95% limit of the population)

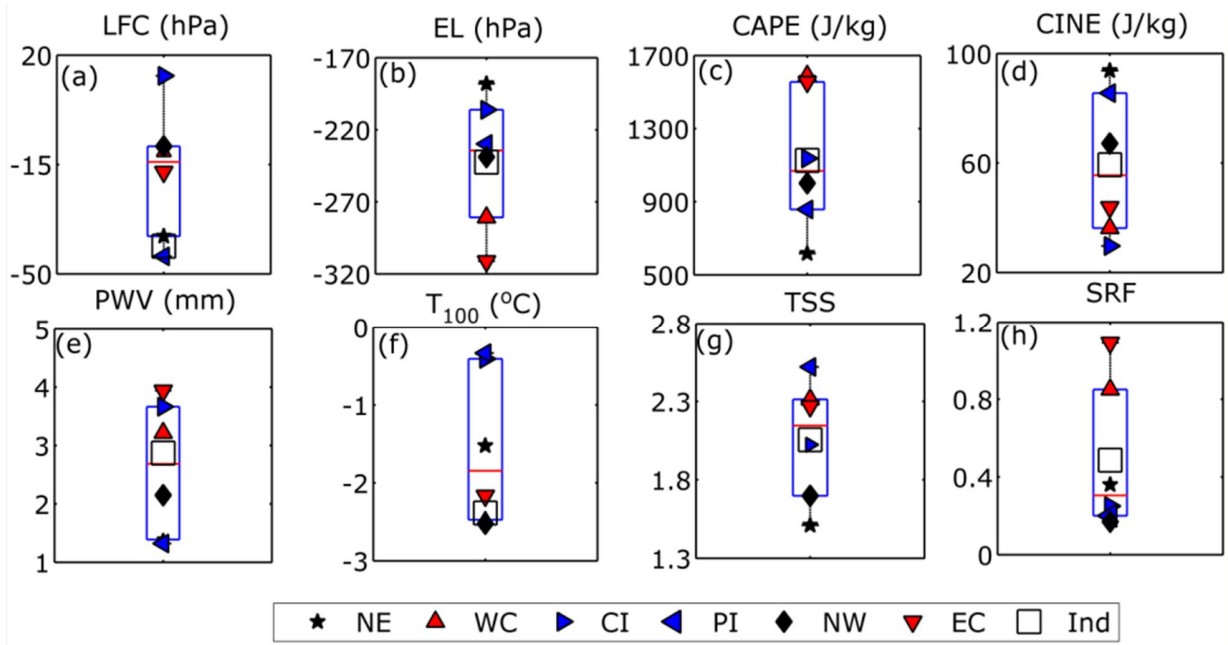

**Figure 5**: Long-term variation of (a) LFC, (b) EL, (c) CAPE, (d) CINE, (e) PWV, (f) T100, (g) TSS and (h) SRF

over the six sub-divisions of India during the period 1980-2016. Legends are same as in Figure 4.

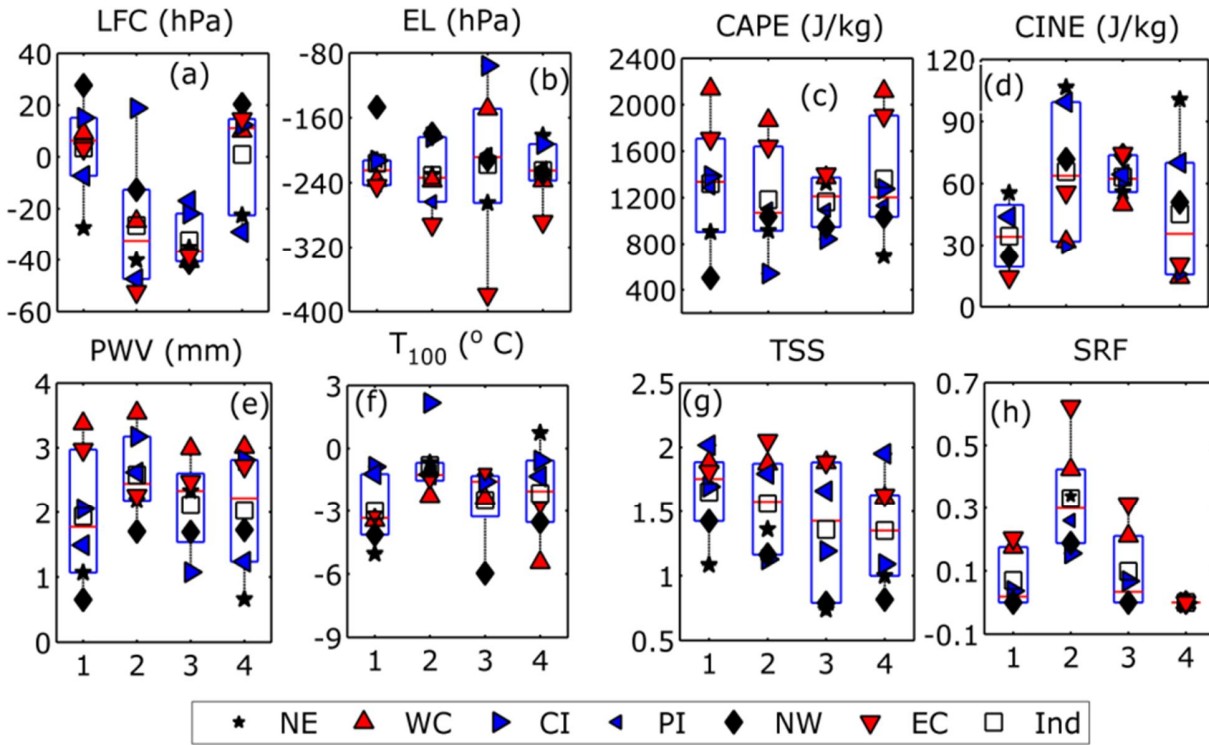

**Figure 6**: Seasonal trend of long-term variation for (a) LFC, (b) EL, (c) CAPE, (d) CINE, (e) PWV, (f) T100, (g) TSS, and (h) SRF over India during all seasons. Here 1 refers to pre-monsoon (March-May), 2 refers to Monsoon (June-September), 3 for Post-monsoon (October-November) and 4 for Winter (December-February). Legends are same as in Figure 4.

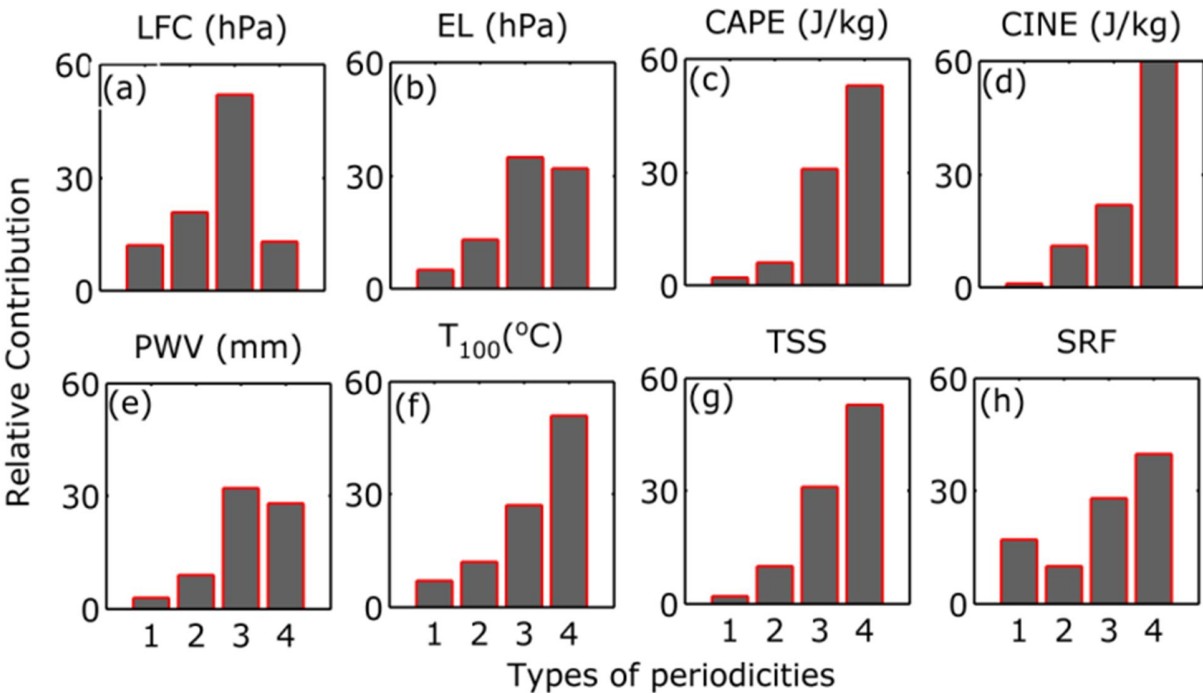

**Figure 7:** Percentage contribution of various periodicities on long-term trend of all instability parameters over India namely: 1.5 -2.5 years periodicity denoted as 1, 4 -6 years periodicity denoted as 2, 10-12 years periodicity displayed as 3 and 16-20 years periodicity represented as 4.

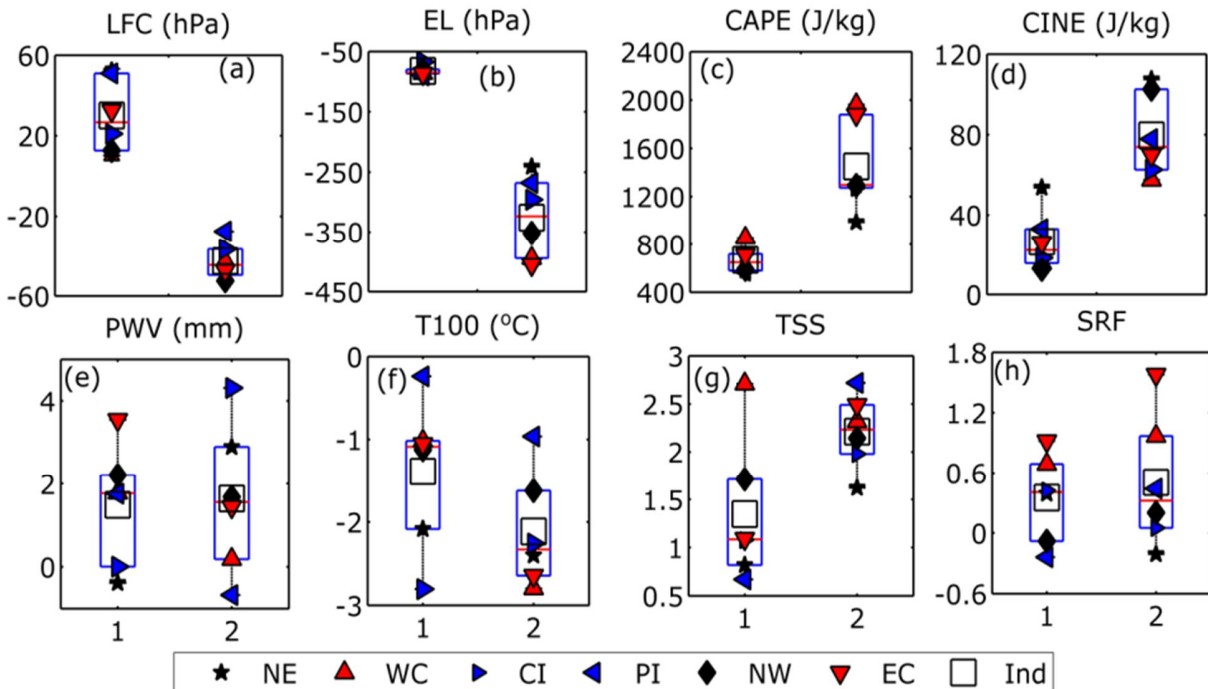

**Figure 8:** Comparison of average values for two time periods indicating the trend of various instability parameter over the six sub-divisions of India in two half periods of 18 years each (the numbers 1 and 2 represent the first and second period, C1 and C2, during 1980-1997 and 1999-2016, respectively) during 1999-2016. Legends are same as in Figure 4.

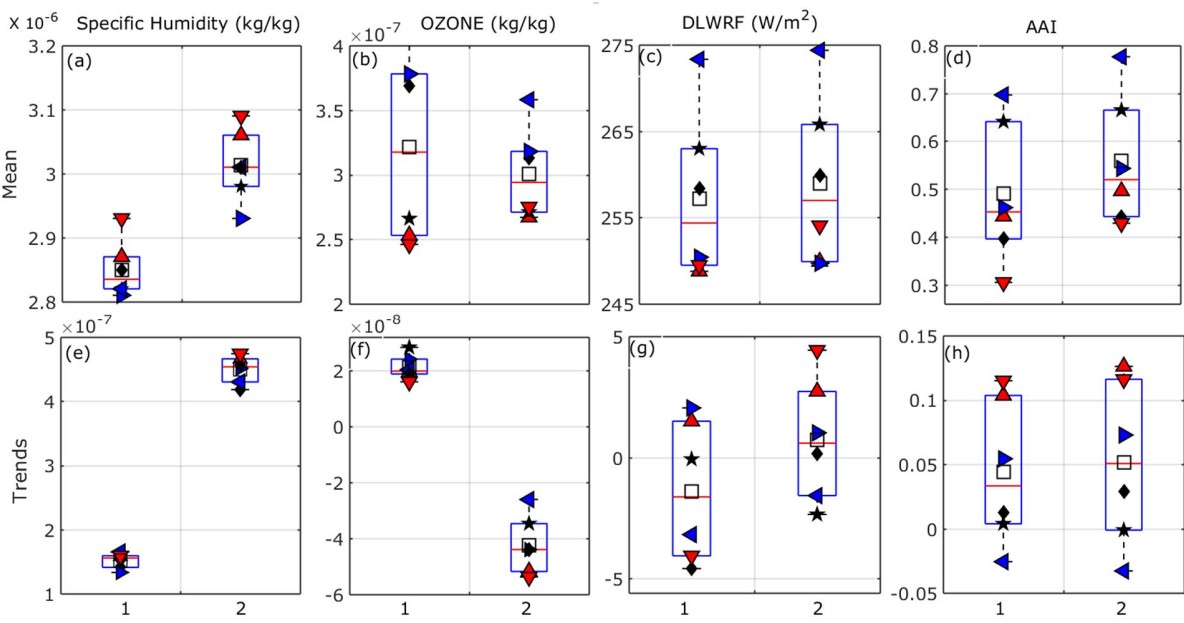

**Figure 9:** Average values and climatological trends of specific humidity, ozone mixing ratio at 100 hPa and Downward Longwave Radiation Flux (DLWRF) with Absorptive Aerosol Index (AAI) over the six sub-divisions of India over two half periods of 18 years each (1980-1997 and 1999-2016). Legends are same as in Figure 4.

# Appendix

**Table A1:** Details of the dataset used.

| Sl no. | Station | Latitude | Longitude | Altitude above MSL | Station Name | Initial No. of profiles available | No. of profiles at 00 & 12Z | No. of profiles at 00Z | Region |
|---|---|---|---|---|---|---|---|---|---|
| 1 | 42361 | 26.23 | 78.25 | 205 | Gwalior | 9901 | 9412 | 4530 | Central India |
| 2 | 42369 | 26.75 | 80.88 | 122 | Lucknow | 16869 | 16387 | 8963 | |
| 3 | 42379 | 26.75 | 83.37 | 78 | Gorakhpur | 12376 | 11793 | 6170 | |
| 4 | 42667 | 23.28 | 77.35 | 522 | Bhopal | 14795 | 13968 | 4472 | |
| 5 | 42809 | 22.65 | 88.45 | 6 | Kolkata | 15212 | 14626 | 6980 | Eastern Coasts |
| 6 | 42971 | 20.25 | 85.83 | 45 | Bhubaneshwar | 18325 | 17552 | 6672 | |
| 7 | 43150 | 17.68 | 83.3 | 70 | Vishakhapatnam | 13225 | 12856 | 6355 | |
| 8 | 43185 | 16.2 | 81.15 | 3 | Machilipatnam | 17108 | 16374 | 8014 | |
| 9 | 43279 | 13 | 80.18 | 14 | Chennai | 14067 | 13487 | 8278 | |
| 10 | 43346 | 10.92 | 79.83 | 7 | Karaikal | 16519 | 16106 | 6890 | |
| 11 | 42314 | 27.48 | 95.02 | 110 | Dibrugarh | 10067 | 9550 | 3801 | North Eastern |
| 12 | 42410 | 26.1 | 91.58 | 54 | Guwahati | 15280 | 14803 | 8681 | |
| 13 | 42492 | 25.6 | 85.17 | 51 | Patna | 8934 | 8318 | 4370 | |
| 14 | 42724 | 23.88 | 91.25 | 16 | Agartala | 15234 | 14732 | 6340 | |
| 15 | 42101 | 30.33 | 76.47 | 251 | Patiala | 11572 | 10129 | 4663 | North Western |
| 16 | 42182 | 28.58 | 77.2 | 210 | New | 14077 | 13982 | 6581 | |
| 17 | 42339 | 26.3 | 73.02 | 217 | Jodhpur | 13133 | 12918 | 5274 | |
| 18 | 42647 | 23.06 | 72.63 | 55 | Ahmadabad | 11430 | 11006 | 5540 | |
| 19 | 42867 | 21.1 | 79.05 | 310 | Sonegaon | 15626 | 14971 | 8532 | Peninsular India |
| 20 | 43014 | 19.85 | 75.4 | 585 | Aurangabad | 14220 | 13993 | 4032 | |
| 21 | 43041 | 19.08 | 82.03 | 554 | Jagdalpur | 10568 | 10205 | 5437 | |
| 22 | 43128 | 17.45 | 78.47 | 530 | Hyderabad | 10234 | 9723 | 6195 | |
| 23 | 43295 | 12.97 | 77.58 | 917 | Bangalore | 10150 | 9514 | 4899 | |
| 24 | 43003 | 19.12 | 72.85 | 14 | Bombay | 14102 | 13808 | 7030 | Western Coasts |
| 25 | 43192 | 15.48 | 73.82 | 58 | Goa | 7070 | 6313 | 5180 | |
| 26 | 43285 | 12.95 | 74.83 | 31 | Mangalore | 9866 | 9406 | 5020 | |
| 27 | 43371 | 8.48 | 76.95 | 60 | Trivandrum | 11590 | 11120 | 8304 | |

**Table A2:** List of Abbreviations

| Slno. | Abbreviation | Full Form |
|---|---|---|
| 1 | LCL (hPa) | Lifted Condensation Level |
| 2 | LFC (hPa) | Level of Free Condensation |
| 3 | EL (hPa) | Equilibrium Level |
| 3 | LI ($^{o}$C) | Lifted Index |
| 4 | VT ($^{o}$C) | Vertical Totals Index |
| 5 | CAPE (J/kg) | Convective Available Potential Energy |
| 6 | MLC (J/kg) | Mixed Layer CAPE |
| 7 | CINE (J/kg) | Convective Inhibition |
| 8 | PWV (mm) | Precipitable Water Vapour |
| 9 | PWL (mm) | Lower Level PWV |
| 10 | WSH ($s^{-1}$) | Wind Shear |
| 11 | T100 ($^{o}$C) | Temperature at 100 hPa |
| 12 | TSO | Ordinary Thunderstorm Frequency |
| 13 | TSS | Severe Thunderstorm Frequency |
| 14 | WRF | Weak Rainfall Frequency |
| 15 | SRF | Severe Rainfall Frequency |
| 16 | SHUM (kg/kg) | Specific humidity |
| 17 | AAI | Absorptive Aerosol Index |
| 18 | IMD | India Meteorological Department |
| 19 | IGRA | Integrated Global Radiosonde Archive |
| 20 | GHG | Green House Gas |
| 21 | DLWRF (W/m$^2$) | Downward Long Wave Radiation Flux |
| 22 | EMD | Empirical Mode Decomposition |
| 23 | UTLS | Upper Troposphere Lower Stratosphere |
| 24 | QBO | Quasi-biennial oscillation |
| 25 | ENSO | El Niño–Southern Oscillation |