# Peer review of "Long-term trends of instability and associated parameters"

_Atmospheric Chemistry and Physics, 2018_

## Referee Comment (RC1) · Anonymous Referee #3 · 17 Oct 2018

Main Comments: Figure 3d: The changes in EL, LI, and CAPE between 1980 and 2016 are difficult to believe and if true compelling. Have other studies shown such huge changes?

Changes in aerosol loading and subsequent changes in the morphology of clouds due to the aerosol indirect effect are not discussed and should be considered when examining trends in stability and precipitation over the Indian region. Please discuss the role of aerosol forcing may play in explaining these trends.

The authors look at trends in 16 different variables derived from radiosonde data. That makes for a difficult read. Might make sense to condense the variables to 8-10 by

removing highly correlated variables.

It is difficult to determine the regional trends from the plots. Perhaps the means/trends for the 3 regions (coastal, interior, and other) can be separated vertically as opposed to stacked on top of each other in plots 4-6 and 8-9.

Specific Comments L36 to L48: I would suggest limiting your references to studies that focused on India. Alternatively, you need to explicitly state for what region and what time period the results you cite are valid.

L54: Increases in air pollution and greenhouse warming may have opposing effects on lower atmospheric stability. Don't group them together here.

L88-90. This sentence is confusing. Are you saying that typically a station has 2-7 gaps with each gap being less than one month in length? If yes, please say so.

Figure 1: Identify the sites with the serial number from Table 1A.

Figure 2: Sufficient space is available at the top of each plot to replace the acronyms with the actual names, e.g., CI –> Central India

Figure 4: This set of plots confuses me. a) By inspecting these plots, is it possible to separate the west coast from the east coast and central India from Peninsula-India? b) Shouldn't there be a separate box and whisker plot for each region? c) I'd suggest flipping the vertical pressure coordinates so that high pressures are located near the bottom and low pressures near the top. d) How can the mean for a region be located at the 5th or 95th percentile? e) Why do I only sometimes see "whiskers"?

L114-116: Is rainfall or lightning required or is the determination strictly based on wind speed?

L135-136: "VT is found to lie exactly in the middle ..." Arguably, TTI or CT is more in the middle than VT.

L163: I do not see a trend in PWL.

L168: "intensification in EL". This is confusing. Go with "increase in the height of the EL".

Table 1: Since the p tests always yield the same results, i.e., significant , I would suggest replacing those columns with columns that indicate the percent trend. I would also suggest adding a column that indicates the units.

L247: "TSS is found to increase drastically". What are the units for TSS. What do you mean by increasing drastically.

L261-262: Chicken and egg question: Is "more convective rain" the cause or consequence of changes in the LFC?

L271: What does it mean for T100 to strengthen?

L290: I find it questionable to look for periodicities of 16-20 years in a data set that is only twice that long.

L318: "Drastic" can mean different things to different people perhaps use a different adjective. Also, be specific as to which instability parameters showed "drastic" changes.

L325: EL –> EL height

L356: ozone breakup –> ozone decreases

L358: cooling effect –> cooling effect due to a reduction in downwelling long wave radiation

L395: What do you mean by "strong cooling due to ozone decomposition?"

L398: Why would the dearth of transported moisture affect that rat of pollutant dispersion by the winds?

L423: CAPE increases in all regions not just near the coast. Please rephrase bullet point 1. Also, "suffer" is a poor choice of words.

L426: "drastic" is a qualitative term - be more quantitative

L439: Are you certain this leads to a strong cooling effect in the troposphere? The increases in OH would lead to increases in the oxidation rates of CO and methane, which could lead to more ozone in the presence of NOx.

TEXT S1: S6: Free condensation –> Free Convection S11: Parcel may continue moving past the EL due to upward momentum. S24: Add a reference to the supercell comment S25-32: "lifted from the LFC to the lowest 100 mb of the troposphere". This is incorrect. Please re-phrase this. I believe the moisture and temperature profiles are averaged over the lowest 100 hPa and then the resulting parcel is lifted to the LCL. S42: calculated as th –> calculated as the S48: resaerch –> research

Minor comments

L32 showed –> shown L39: due to surface heating –> due to increases in surface heating L46: extreme precipations –> extreme precipitation events L46: intense convections –> intense convection L54: lower instability is reducing –> lower tropospheric instability is decreasing L58: studies over India has –> studies over India have L108: upto –> up to L171: reduction VT –> reduction in VT L192: higher in the coasts –> higher at the coasts L195: "higher", Do you mean "more negative?" L217: "ttset" –> test L223: "all the regions" –> not true in the NE region L224: is minimum –> is smallest L227: "also show an enhancement " –> become more negative L243-244: Smallest changes in the NE and NW regions. The difference between inland and coast regions isn't that large (2 versus 2.375 degrees) L376: a dominant increase –> an increase L380: resulting more –> resulting in more L382: To prove this hypothesis –> To test this hypothesis L383: increasing prominently –> increasing L383: expand DLWRF acronym
* * *

---

## Referee Comment (RC2) · Anonymous Referee #1 · 14 Nov 2018

General comments:

This manuscript investigates a comprehensive data set of radiosonde measurements over India (1980-2016) by calculating a number of different instability parameters and other relevant thunderstorm indices (totally 16 parameters). For six different regions in India the climatological mean values, long-term variations and their seasonal trends, periodicities, and long-term trends of all these parameters are presented. The latter indicates strong enhancement of Convective Available Potential Energy (CAPE) and the intensification of severe thunderstorm occurrence in coastal regions caused by the enhanced vertical moisture transport (due to increase in Hadley cell and Brewer-

[Figure]

Dobson circulation strength) and an associated cooling at 100 hPa. Similar results for long-term trends over India have been reported in previous publications, however in those studies mainly CAPE was studied and not a variety of other parameters as in this study.

The topic and the general length of the paper are relevant for publication in ACP. The introductory discussion is adequate and the referencing is sufficient. Generally the paper is well structured and logical, however sometimes single sentences are difficult to follow when the figures are described in detail (for recommendations see below). At the end of some chapters the main findings are well summarised which simplifies the reading. Also the discussion in chapter 4 is well written. One main weakness of the manuscript is the lack of instrument description (type of radiosondes and sensors) and discussion about changes in the instrumentation between the years 1980-2016, that might cause trends. Furthermore, the methods used are not described in detail and not sufficient references are given, e.g. for the calculated instability indices (for improvement see below). A high number of the given references in the text are spelled different in the reference list. Some of the figures and legends need improvement. In summary the manuscript contains many minor mistakes, parts of the text need substantial improvement, however the data set is comprehensive, a sufficient number of parameters were calculated and the trend analyses are interesting and important.

For the reasons mentioned above and below the paper is appropriate for publication in ACP after a major revision.

Specific comments:

For the description of the results in the figures where the pressure (in hPa) is shown along the y-axis (Figs. 3, 4, 5, 6, 8 (a, b, c)), it is recommended to improve the wording in the text to simplify the reading (especially the pages 5-9). First describe what you see in the figure: increasing or decreasing pressure and then describe what it means for the altitude (opposite direction as pressure changes). Always add if you describe

pressure or height to avoid too much confusion during this presentation. Also CINE is a parameter that has negative values and when you write CINE strengthens/enhance it means that values are more negative. These type of statements are confusing for the reader. Instead of describing the parameters in the order of the figures (a, b, c,…) try to describe all parameters that indicate more stability together and all parameters that more instability together (the same also for increasing/decreasing heights) and then try to interpret what this observation means.

The complete text on the pages 5-9 needs a major revision in this direction. The paper contains a high number of such confusing sentences. Just one example of this type of confusing sentences is Page 9, Line 326-328: "LI is expected to strengthen from 37 years trend, however it shows a slight weakening in C1 followed by a prominent strengthening in C2 resulting in a net increasing trend". Instead of weakening/strengthen use the expressions "more stable/less stable" or use "instable". You also have to add OF WHAT you observe an increasing trend. Contrarily to these sentences that are hard to read, your summaries at the end of the paragraphs with the detailed descriptions are in a good shape and perhaps always use italic style to single them out.

Further examples of strange wording are expressions like e.g. "two-part trend/analysis".

Another example on page 11, line 426-428: "Seasonal variation of LFC, CINE, Wind Shear (WSH), TSO and WRF shows drastic increase during monsoon and post-monsoon seasons while strengthening in CAPE, EL, Lifted Index (LI) and TSS are found more prominent during the pre-monsoon." Instead write: in the pre-monsoon increasing TSS activity is observed due to higher instability connected to increasing EL height and CAPE values, decreasing LI values and so on.

Minor comments and technical corrections:

Page 1, line 18: Replace "the increase in TSS, SRF and CAPE is found more severe

after the year 1999" by "the increase in TSS and CAPE is stronger after the year 1999". Cut SRF because it is not so strong. Here you already mention CAPE and then first in line 20 you write Convective Available Potential Energy, improve.

Page 1, line 33: Always cite references chronologically (check throughout manuscript).

Page 1, line 35: Change to "Huntrieser et al." Check throughout manuscript if all references, where needed, have "et al.".

Page 2, line 38: Replace "has" by "have".

Page 2, line 45: Replace by "…ingress. Consequently the…"

Page 2, line 48: Replace "2016" by "2017". Compare the year of all references in the text with the year in the reference list.

Page 2, line 49: Replace "Over Indian region" by "Over the Indian region". Check throughout manuscript.

Page 2, line 63: Replace "is" by "was".

Page 2, line 66: Replace "Thunder Storm" by "Thunderstorm". Check throughout manuscript.

Page 2, line 77: Also add web link to the archive.

Page 3, line 79: "Zhe et al., 2013" is missing in reference list.

Page 3, line 91: Replace "years X 12" by "years x 12".

Page 3, line 107: Replace "in Supplementary" by "in the Supplementary".

Page 3, line 114: Give link to the website.

Page 3 bottom and Page 4 top: Here you introduce your 14 parameters. Also add your abbreviation for wind shear, temperature, severe and weak rainfall days.

Page 4, line 127-136: PCA is suddenly introduced. Also add a reference to this method

except your own reference. Better explain how the methods work, why you use it here and what you show in Fig. 2 (paragraph needs improvement). Your argumentation to reduce the parameters to LI and VT only is not clear to me.

Page 4, line 143. Give references or web link to the data set.

Page 5, line 161: Give coordinates and location of Gadanki.

Page 5, line 187: Here you for the first time mention boxplot analysis. Give a reference and better to introduce all types of analysis methods in Chapter 2.

Page 5, line 192: Replace "higher in the coasts" by "higher in coastal areas".

Page 6, line 217: Here you for the first time mention "ttset analysis". Replace by "t-test analysis". Add a reference and introduce all types of analysis methods in Chapter 2.

Page 7, line 260: Replace "The seasonal variation of atmospheric instability" by "The seasonal variation of the long-term variations of atmospheric instability".

Page 7, line 275: Replace "donot" by "do not".

Page 8, line 290: What is the 16-20 years periodicity related to?

Page 9, line 317: Replace "Investigation of two-part trends" by "Trend investigations: 1980-1997 and 1999-2016"

Page 9, line 321: Replace "segments" by "time periods"

Page 9, line 351: "ascent in EL". Just another example of expressions that might cause confusion. In the figure you show EL with pressure. Here you better add "ascent in EL height/altitude".

Page 10, line 362: What is the reason for the net increase in the Hadley cell?

Page 10, line 368: The expression "two-part trend" is strange. Replace it throughout the manuscript.

Page 10, line 380: Replace "earth" by "Earth".

Page 11, line 398-399: Strange wording "the lower atmospheric instability gets limited in inland regions".

Page 11, line 401: Here you write "the trend in AAI are not significantly different in the two halves of the analysis". Replace by "the trend in AAI is not significantly different for the two time periods C1 and C2". This sentence is also contrary to your sentence on page 10, line 385, where you write "the mean of AAI is increasing sharply in C2 with a positive trend".

Page 11, line 411: Replace "check" by study/analyse.

Page 11, line 415: Add the years.

Page 11, line 432: Not all parameters show a rise.

Page 11, line 435: Add also that there might be an influence from pollution.

Page 12, line 445: Replace "EL has" by "EL height increase has".

References: A high number of the given references in the text are spelled different in the reference list, here just some examples from the text (check all your references thoroughly): Annanthakrishnan, Shanti, Reimann-Campe, Trenbreth, Murthy and Shivaramakrishnan, Allapattu and Kunnikrishnan, Mohankumar, Anderson.

Page 12, Line 469: Replace "Res.." by "Res."

Page 12, Line 472: Reference is not detailed enough to find. Give web link.

Page 13, Line 478: Add blank before 2016.

Page 13, Line 481: Chakraborty is a new reference. Put below.

Page 13, line 491-492: Check the writing of degree.

Page 13, line 502: Separate: "thestrengthening"

Page 13, line 508: Put "Huntrieser" after "Guha". Replace "traditional" by "Traditional".

Page 14, line 522: Replace "Past" by "past".

Page 14, Line 532-533: Replace "Radiosonde" by "radiosonde". Add "a" after "2018".

Page 14, Line 536: Add "b" after "2018".

Page 14, Line 538: Check symbols around numbers.

Page 14, Line 548: "Raipal" is wrong: The authors are "Joseph, P.V., Raipal, D. K., Deka, S. N.. Page 15, line 563: Separate: "thestrengthening"

Page 15, line 575: Add blank ahead of "2011".

Table 1: Add to the legend what your symbols mean (micro: average, sigma: standard deviation, p: significance from t-test). Write "Information" with a small letter.

Fig. 1: Legend text: "NC" replace by "CI" (in the map)

Fig. 3: Long-term variations shown (not averages?), how do you calculate this, why are values negative? Why is the CAPE (also e.g. WSH) variation so small 1980-1990 and then so large? Is the radiosonde type always the same?

Fig. 4: For a better overview I recommend to insert the used symbols in one of the figures (e.g. upper right or left figure) and also write for which regions (CI, PI, NE, EC, WC, NW) which symbols are used (replicate in Fig. 5, 6, 8, 9). In the print-out it is difficult to separate the blue and black symbols. Replace blue by white or yellow symbols. Add in the legend text that the red line is the median value. What is the smaller box in the bigger box? In Fig. (h) the smaller box is missing. In the header of Fig. (d) there is a blue small square, cut. The same square is also present in Figs. 5d, 6d and 8d.

Fig. 5: The left axis of Fig. (e) is partly cut, it is not visible anymore that the values are negative.

Fig. 6: In two of the headers (d) and (l) there is a blank between ° and C. In the legend text you have to add that you show "the seasonal trend of the long-term variation shown in Fig. 5". Write Monsoon, Post, Winter with small letters. A dot is missing at the end of the sentence (December-February).

Fig. 7: Cut the blanks between the years and use the same "-" sign.

Fig. 8: Replace "Average long-term trends" by "Comparison of average values for two time periods indicating the trend of various instability parameter. . .". Replace one sentence in the legend by "(the numbers 1 and 2 represent the first and second period, C1 and C2, during 1980-1997 and 1999-2016, respectively)".

Fig. 9: Change to "Average values (mean)" in the legend. Change to "Downward Longwave Radiation Flux (DLWRF) and Absorptive Aerosol Index (AAI)".

Table A1: Add also the height of the stations.

Table A2: Try to add a reference to each of the indices that are based on equations e.g. like CAPE. The reader must be able to also calculate the same parameters.

---

## Referee Comment (RC3) · Anonymous Referee #2 · 15 Nov 2018

This paper estimates the atmospheric stability parameters from the radiosonde data over the Indian sub-continent (sub classified to six regions) and discusses the long-term trends. From meteorological point of view, the analysis of atmospheric stability parameter is important in examining the convective weather development. The manuscript is of interest, however needs major revision and careful consideration of the issues listed below.

Ln 40: projected a 236 % increase. Increase in a year/decade?

Ln 50: strong relationship. Whether the relationship is positive or negative. Elaborate.

Ln 76: How the homogeneity and the quality of the radiosonde data have been as

assessed. How the authors have taken care in raining condition or when there is satu-ration in humidity measurement?

Ln 96: In 37 years of data, how number of profile is more than ~13514 (if there is one profile per day)

Lnb 104: Any reason for choosing cubic spline interpolation? Why not linear ?

Ln 105: Calculations of LCL, LFC, EL, CAPE, MLC, CINE etc are the key points of your manuscript. Please mention the formulas in the manuscript. Radiosonde measure-ment depends on balloon burst altitude. Is the calculation of instability parameters is performed for available radiosonde measurement height or the authors have restricted their analysis when the data is available up to minimum height level (like 20 km). Please discuss those points in detail for each station?

Ln 113: Do the radiosonde measure surface wind? How reliable is the surface wind data?

Ln131: Why the PCA is performed on the yearly data? Do the analysis is performed for all days or only for TSS and TSO days?

Ln 160: Why the authors need to compare with Chennai? Also couldn't find any com-parison dataset? Is this monthly/yearly mean data? What the error bar describe? Why TSO, TSS, WRF and SRF don't have error bar? How trend is calculated? What is the slope value? What information one can extract from such trends? What the positive and negative value means? What is the meaning of increasing CAPE in the atmosphere?

Ln 168: What height?

Ln169: reduction in temperatures near 100 hPa (Fig.3l) plays an important role in modulating the total atmospheric instability and CAPE. Why 100 hPa? Do the authors have any hypothesis to demonstrate it?

[Figure]

Ln217: What is ttset analysis?

Ln241: Hence, the mid and upper tropospheric moisture plays a crucial role in modulating the Indian climate. This sentence is not clear.

Ln290: You dataset is for 37 years, how you can get periodicity up to 20 years?

Ln319: trends before and after the period 1996-2000 are significantly different from each other. Any reason? Any change in radiosonde sensor?

———————————————————

---

## Author Response (AR1)

**Replies to Reviewer #1 Comments/suggestions**

First of all we wish to thank the reviewer for providing constructive comments/suggestions which significantly improved the content of the manuscript. The authors have addressed all the comments raised by the three reviewers and incorporated in the revised manuscript.

General comments:

This manuscript investigates a comprehensive data set of radiosonde measurements over India (1980-2016) by calculating a number of different instability parameters and other relevant thunderstorm indices (totally 16 parameters). For six different regions in India the climatological mean values, long-term variations and their seasonal trends, periodicities, and long-term trends of all these parameters are presented. The latter indicates strong enhancement of Convective Available Potential Energy (CAPE) and the intensification of severe thunderstorm occurrence in coastal regions caused by the enhanced vertical moisture transport (due to increase in Hadley cell and Brewer Dobson circulation strength) and an associated cooling at 100 hPa. Similar results for long-term trends over India have been reported in previous publications, however in those studies mainly CAPE was studied and not a variety of other parameters as in this study.

The topic and the general length of the paper are relevant for publication in ACP. The introductory discussion is adequate and the referencing is sufficient. Generally the paper is well structured and logical, however sometimes single sentences are difficult to follow when the figures are described in detail (for recommendations see below). At the end of some chapters the main findings are well summarised which simplifies the reading. Also the discussion in chapter 4 is well written. One main weakness of the manuscript is the lack of instrument description (type of radiosondes and sensors) and discussion about changes in the instrumentation between the years 1980-2016, that might cause trends. Furthermore, the methods used are not described in detail and not sufficient references are given, e.g. for the calculated instability indices (for improvement see below). A high number of the given references in the text are spelled different in the reference list. Some of the figures and legends need improvement. In summary the manuscript contains many minor mistakes, parts of the text need substantial improvement, however the data set is comprehensive, a sufficient number of parameters were calculated and the trend analyses are interesting and important.

Reply: The authors thank the reviewer for appreciating actual content of the work and providing constructive comments/suggestions which significantly improved the content of the manuscript. The authors have addressed all the comments raised by the reviewer and incorporated in the revised manuscript.

Specific comments:

For the description of the results in the figures where the pressure (in hPa) is shown along the y-axis (Figs. 3, 4, 5, 6, 8 (a, b, c)), it is recommended to improve the wording in the text to simplify the reading (especially the pages 5-9). First describe what you see in the figure: increasing or decreasing pressure and then describe what it means for the altitude (opposite direction as pressure changes). Always add if you describe pressure or height to avoid too much confusion during this presentation. Also CINE is a parameter that has negative values and when you write CINE strengthens/enhance it means that values are more negative. These type of statements are

confusing for the reader. Instead of describing the parameters in the order of the figures (a, b, c,..) try to describe all parameters that indicate more stability together and all parameters that more instability together(the same also for increasing/decreasing heights)and then try to interpret what this observation means.

The complete text on the pages 5-9 needs a major revision in this direction. The paper contains a high number of such confusing sentences. Just one example of this type of confusing sentences is Page 9, Line 326-328: "LI is expected to strengthen from 37 years trend, however it shows a slight weakening in C1 followed by a prominent strengthening in C2 resulting in a net increasing trend". Instead of weakening/strengthen use the expressions "more stable/less stable" or use "instable". You also have to add OF WHAT you observe an increasing trend. Contrarily to these sentences that are hard to read, your summaries at the end of the paragraphs with the detailed descriptions are in a good shape and perhaps always use italic style to single them out.

Reply: The authors have gone through the figure descriptions in pages 5-9 and corrected all the confusing sentences. Words like weakening/ strengthening have been replaced with increase/decrease of magnitude/ value and also by more/less unstable. The authors have also taken care that they mention prominently which parameter change is being referred to in all the areas of the manuscript. The following types of changes have been done:

Instead of increase/decrease the terms ascend/descend has been used for EL/LFC/LCL to get rid of the height-pressure confusion in readers. Next, the increase/decrease in magnitude of LI/CINE is used in place of strengthening/ weakening in the manuscript. Strengthening/Weakening of T100 has been replaced by more/less cooling in 100 hPa. Finally, Strengthening or weakening of others like CAPE, MLC, VT, PWV, PWL, TSO, TSS, WRF and SRF has been be rewritten as increasing or decreasing in the manuscript and in the first mention, it has also been cleared that its increase has strengthened the instability.

This comment (italic style) from the reviewer cannot be implemented to abide the ACP text formatting rules.

Further examples of strange wording are expressions like e.g. "two-part trend/analysis".

Reply: As each half of the time series is about 18 years which is close to two decades, hence the two-part trend has been rewritten as quasi-bi-decadal trends in the manuscript.

Another example on page 11, line 426-428: "Seasonal variation of LFC, CINE, Wind Shear (WSH), TSO and WRF shows drastic increase during monsoon and postmonsoon seasons while strengthening in CAPE, EL, Lifted Index (LI) and TSS are found more prominent during the pre-monsoon." Instead write: in the pre-monsoon increasing TSS activity is observed due to higher instability connected to increasing EL height and CAPE values, decreasing LI values and so on.

Reply: As per reviewer suggestion we have incorporated these sentences.

Minor comments and technical corrections:

Page 1, line 18: Replace "the increase in TSS, SRF and CAPE is found more severe C3 after the year 1999"by"the increase in TSS and CAPE is stronger after the year 1999". Cut SRF because it is not so strong. Here you already mention CAPE and then first in line 20 you write Convective Available Potential Energy, improve. Page 1, line 33: Always cite references chronologically (check throughout manuscript).

Page 1, line 35: Change to "Huntrieser et al." Check throughout manuscript if all references, where needed, have "et al.".

Page 2, line 38: Replace "has" by "have".

Page 2, line 45: Replace by "...ingress. Consequently the..." Page 2, line 48: Replace "2016" by "2017". Compare the year of all references in the text with the year in the reference list.

Page 2, line 49: Replace "Over Indian region" by "Over the Indian region". Check throughout manuscript. Page 2, line 63: Replace "is" by "was".

Page 2, line 66: Replace "Thunder Storm" by "Thunderstorm". Check throughout manuscript.

Page 2, line 77: Also add web link to the archive.

Reply: These corrections have been done in the revised manuscript.

Page 3, line 79: "Zhe et al., 2013" is missing in reference list.

Reply: The authors could not locate Zhe et al. 2013, hence they have introduced a new and more detailed reference on IGRA Data named Ferreira et al., 2018.

Page 3, line 91: Replace "years X 12" by "years x 12". Page 3, line 107: Replace "in Supplementary" by "in the Supplementary". Page 3, line 114: Give link to the website.

Reply: All these corrections have been done in the revised manuscript.

Page 3 bottom and Page 4 top: Here you introduce your 14 parameters. Also add your abbreviation for wind shear, temperature, severe and weak rainfall days.

Reply: Here, the mentioned 14 parameters refer to the parcel and moisture parameters along with severe frequency estimates. Other than this, 8 standard instability parameters should still be considered for analysis. However, it is known that most of these instability parameters are inter-related; hence PCA analysis is done to identify and use only those instability parameters that can give a complete but independent overview of the atmospheric instability using minimum parameters. The PCA analysis revealed that LI and VT are the most important instability parameters affecting thunderstorms, hence these two are added with the previous set of 14 parameters to get total 16 parameters in this study.

The rainfall frequency abbreviations have been added here, however the other two could not be incorporated as those parameters are not mentioned there.

Page4,line127-136: PCA is suddenly introduced. Also add are ference to this method except your own reference. Better explain how the methods work, why you use it here and what you show in Fig. 2 (paragraph needs improvement). Your argumentation to reduce the parameters to LI and VT only is not clear to me.

Reply: To get rid of the confusion why PCA is used, what is it and what it reveals, the following has been added in the revised manuscript:

*From the previous section it follows that a set of 14 parcel parameters with rainfall and thunderstorm frequencies are essential to understand the convective climatology over India. However, other than this, 8 standard instability parameters (LI, KI, TTI, CT, VT, CAPE, CINE and MLC) are also additionally important to quantify the thunderstorm severity, hence must also be considered for analysis. Now, it is known that most of these instability parameters are inter-related; hence PCA analysis is done to identify and use only those instability parameters that can give a complete but independent overview of the atmospheric instability using minimum*

*parameters. In this analysis, introduced by (Hoteling 1936) a set of possibly related parameters are converted into orthogonal independent components after which the primary components are plotted with the initial parameters. Parameter variance scores present at the farthest distance from the primary principal components and also from all the other variables contain the highest variance; hence they are selected for representing the existing group of old inputs. Consequently, the analysis revealed that LI and VT depict the highest variance from the other six instability parameters; hence these two are added with the previous set of 14 parameters to get total 16 parameters in this study.*

Page 4, line 143. Give references or web link to the data set.

Reply: This correction has been done in the revised manuscript.

Page 5, line 161: Give coordinates and location of Gadanki.

Reply: This correction has been done in the revised manuscript as follows:

*In the previous study by Chakraborty et al. (2018), long term trends of instability were investigated over Gadanki($13.5^oN, 79.2^oE$) situated on a hilly terrain with an altitude of 370 m above sea level at a distance of ~150 km from the eastern coasts and Bay of Bengal*

Page 5, line 187: Here you for the first time mention boxplot analysis. Give a reference and better to introduce all types of analysis methods in Chapter 2.

Reply: This correction has been done in the revised manuscript.

Page 5, line 192: Replace "higher in the coasts" by "higher in coastal areas".

Reply: This correction has been done in the revised manuscript.

Page 6, line 217: Here you for the first time mention "ttset analysis". Replace by "t-test analysis". Add a reference and introduce all types of analysis methods in Chapter 2.

Reply: This correction has been done in the revised manuscript.

Page 7, line 260: Replace "The seasonal variation of atmospheric instability" by "The seasonal variation of the long-term variations of atmospheric instability".

Reply: This correction has been done in the revised manuscript.

Page 7, line 275: Replace "donot" by "do not".

Reply: This correction has been done in the revised manuscript.

Page 8, line 290: What is the 16-20 years periodicity related to?

Reply: The authors would like to clarify that this 16-20 year periodicity is not found related to any known natural phenomena according to existing research knowledge. This periodicity has been the outcome of statistical analysis and it may simply be a complex harmonic combination of 11 years and 5 years periodicity from the solar effect and ENSO phenomena. But in spite of having no strong physical explanation, the changes in the trends of instability and related parameters are mostly found prominent between the years 1996-2004. Hence, the relative contribution of this periodicity appears to be more significant than the others. Nevertheless, in future, subject to more data availability, this periodicity can be investigated in further detail to address this issue.

Page 9, line 317: Replace "Investigation of two-part trends" by "Trend investigations: 1980-1997 and 1999-2016"

Reply: In view of comments from other reviewer, the sub-section title has been kept as "Investigation of quasi-bi-decadal trends between 1980-1997 and 1999-2016".

Page 9, line 321: Replace "segments" by "time periods"

Reply: This correction has been done in the revised manuscript.

Page 9, line 351: "ascent in EL". Just another example of expressions that might cause confusion. In the figure you show EL with pressure. Here you better add "ascent in EL height/altitude".

Reply: The authors consider that EL is an imaginary layer in the atmosphere, hence ascent of EL is self-explanatory.

Page 10, line 362: What is the reason for the net increase in the Hadley cell?

Reply: Recent studies have found an association between Hadley Cell circulation intensification and global warming induced by green house gases. However, this hypothesis has not yet been proved till date and hence this part will not be incorporated in the revised manuscript.

Page 10, line 368: The expression "two-part trend" is strange. Replace it throughout the manuscript.

Reply: This correction has been done in the revised manuscript.

Page 10, line 380: Replace "earth" by "Earth".

Reply: This correction has been done in the revised manuscript.

Page 11, line 398-399: Strange wording "the lower atmospheric instability gets limited in inland regions".

Reply: This line has been modified as "As a result, the growth of lower atmospheric instability gets inhibited in the inland regions."

Page 11, line 401: Here you write "the trend in AAI are not significantly different in the two halves of the analysis". Replace by "the trend in AAI is not significantly different for the two time periods C1 and C2". This sentence is also contrary to your sentence on page 10, line 385, where you write "the mean of AAI is increasing sharply in C2 with a positive trend".

Reply: All these corrections have been done in the revised manuscript, the word sharply has been replaced by slightly in accordance with the later line.

Page 11, line 411: Replace "check" by study/analyse.

Reply: Replaced.

Page 11, line 415: Add the years.

Reply: Added.

Page 11, line 432: Not all parameters show a rise.

Reply: Corrected.

Page 11, line 435: Add also that there might be an influence from pollution.

Reply: Added.

Page 12, line 445: Replace "EL has" by "EL height increase has".

Reply: Replaced.

References: A high number of the given references in the text are spelled different in the reference list, here just some examples from the text (check all your references thoroughly): Annanthakrishnan, Shanti, Reimann-Campe, Trenbreth, Murthy and Shivaramakrishnan, Allapattu and Kunnikrishnan, Mohankumar, Anderson.

Reply: All these corrections have been done in the revised manuscript.

Page 12, Line 469: Replace "Res.." by "Res."

Reply: Replaced.

Page 12, Line 472: Reference is not detailed enough to find. Give web link.

Reply: Provided detailed reference.

Page 13, Line 478: Add blank before 2016. Page 13,

Reply: Added.

Line 481: Chakraborty is a new reference. Put below.

Reply: Corrected.

Page 13, line 491-492: Check the writing of degree.

Reply: Corrected.

Page 13, line 502: Separate: "thestrengthening" C6

Reply: Separated.

Page 13, line 508: Put "Huntrieser" after "Guha". Replace "traditional" by "Traditional".

Reply: Replaced.

Page 14, line 522: Replace "Past" by "past".

Reply: Replaced.

Page 14, Line 532-533: Replace "Radiosonde" by "radiosonde". Add "a" after "2018".

Reply: This correction has been done in the revised manuscript.

Page 14, Line 536: Add "b" after "2018".

Reply: Added.

Page 14, Line 538: Check symbols around numbers.

Reply: Corrected.

Page 14, Line 548: "Raipal" is wrong: The authors are "Joseph, P.V., Raipal, D. K., Deka, S. N.. Page 15, line 563: Separate: "thestrengthening"

Reply: This correction has been done in the revised manuscript.

Page 15, line 575: Add blank ahead of "2011".

Reply: Added.

Table 1: Add to the legend what your symbols mean (micro: average, sigma: standard deviation, p: significance from t-test). Write "Information" with a small letter.

Reply: Added.

Fig. 1: Legend text: "NC" replace by "CI" (in the map)

Reply: Replaced.

Fig. 3: Long-term variations shown (not averages?), how do you calculate this, why are values negative? Why is the CAPE (also e.g. WSH) variation so small 1980-1990 and then so large? Is the radiosonde type always the same?

Reply: These are the 37 year annual average plots of all parameters calculated by robust-fit technique.

The values have been normalized with respect to mean and hence they appear negative.

CAPE, TSO, EL, SRF and WSH show sudden increase in annual regional averages. This cannot be due to change in radiosonde type, because TSO, WSH and SRF are utilized from outside the IGRA database.

IMD radiosonde stations use IM-MK3 radiosonde which utilizes thermistor for temperature and carbon hygristor for humidity sensing all throughout its operational life. More details are not mentioned on the IMD data quality change but the type of radiosonde did not change in the last few decades.

Fig. 4: For a better overview I recommend to insert the used symbols in one of the figures (e.g. upper right or left figure) and also write for which regions (CI, PI, NE, EC, WC, NW) which symbols are used (replicate in Fig. 5, 6, 8, 9). In the print-out it is difficult to separate the blue and black symbols. Replace blue by white or yellow symbols. Add in the legend text that the red line is the median value. What is the smaller box in the bigger box? In Fig. (h) the smaller box is missing. In the header of Fig. (d) there is a blue small square, cut. The same square is also present in Figs. 5d, 6d and 8d.

Reply: Symbol legends have now been added in the figure. To preserve parity among the marker sizes, the size of symbols could not be increased drastically in any sub-division. Hence, use of separate colouring has not been done. There is only one square box possible in each subplot. The sizes differ due to manual type editing errors during figure finalization.

Fig. 5: The left axis of Fig. (e) is partly cut, it is not visible anymore that the values are negative.

Reply: This correction has been done in the revised manuscript.

Fig. 6: In two of the headers (d) and (l) there is a blank between ∘ and C. In the legend textyouhavetoaddthatyoushow"theseasonaltrendofthelong-termvariationshown in Fig. 5". Write Monsoon, Post, Winter with small letters. A dot is missing at the end of the sentence (December-February).

Reply: All These corrections have been done in the revised manuscript.

Fig. 7: Cut the blanks between the years and use the same "-" sign.

Reply: This correction has been done in the revised manuscript.

Fig. 8: Replace "Average long-term trends" by "Comparison of average values for two time periods indicating the trend of various instability parameter...". Replace one sentence in the

legend by "(the numbers 1 and 2 represent the first and second period, C1 and C2, during 1980-1997 and 1999-2016, respectively)".

Reply: Replaced.

Fig. 9: Change to "Average values (mean)" in the legend. Change to "Downward Longwave Radiation Flux (DLWRF) and Absorptive Aerosol Index (AAI)".

Reply: All These corrections have been done in the revised manuscript.

Table A1: Add also the height of the stations.

Reply: Added.

Table A2: Try to add a reference to each of the indices that are based on equations e.g. like CAPE. The reader must be able to also calculate the same parameters.

Reply: All These corrections have been done in the revised supplementary section.

We thank the reviewer once again for providing detailed suggestions which made us to improve the manuscript content.
* * *
**Replies to Reviewer #2 Comments/suggestions**

This paper estimates the atmospheric stability parameters from the radiosonde data over the Indian sub-continent (sub classified to six regions) and discusses the longterm trends. From meteorological point of view, the analysis of atmospheric stability parameter is important in examining the convective weather development. The manuscript is of interest, however needs major revision and careful consideration of the issues listed below.

Reply: First of all we wish to thank the reviewer for providing constructive comments/suggestions which significantly improved the content of the manuscript. The authors have addressed all the comments raised by the three reviewers and incorporated in the revised manuscript.

Ln 40: projected a 236 % increase. Increase in a year/decade?

Reply: This 236% increase indicates the total projected rise from the decade 1980-1990 to 2080-2090 over Eastern United States.

Ln 50: strong relationship. Whether the relationship is positive or negative. Elaborate.

Reply: A prominent negative relationship was revealed between 100 hPa temperature and CAPE. Temperatures at 100 hPa had shown heating between 1980 and 1989 and then cooling while CAPE decreased first and then it increased after 1990. This relationship was found more prominent when the trends of T100 and CAPE were associated on a seasonal basis and the periodicity of T100 also matched with the solar cycle variations. Hence solar activity and convection was thought responsible for the temperature changes at 100 hPa. This has now been briefly added in the revised manuscript as follows.

*Dhaka et al. (2010) utilized radiosonde observations during 1958-1997 and obtained very prominent anti-correlations on both yearly and seasonal basis between convection strength (CAPE) and upper troposphere temperatures at 100 hPa (T100).*

Ln 76: How the homogeneity and the quality of the radiosonde data have been as C1 assessed. How the authors have taken care in raining condition or when there is saturation in humidity measurement?

Reply: The authors have taken care of all three issues pertaining to data quality namely: volume, instrument type and quality. First, in majority of the IGRA stations, number of utilized profiles is more than 5000 which is quite a descent population size to calculate 37 year trends. Additionally, the number of profiles in every station are found to be almost homogenously distributed with respect to years and seasons (except monsoon). Secondly, data of all Indian IGRA stations come from a single type of IM-MK3 radiosondes which has not undergone any significant change in radiosonde accuracies in the last years; thereby cancelling the chances of sudden instrument bias. Finally, in case of quality, the authors have used their own certain quality check procedures other than 7 additional quality checks already done by the IGRA database before accepting the data; hence this is expected to address all data homogeneity issues in the utilized data.

Since the work itself is on thunderstorms and rainfall, hence, raining conditions had to be taken in the datasets. However, if the profiles do have any saturation in humidity measurements then those will be rejected by the value repetition quality check of IGRA(Durre et al., 2006). Hence those errors may not impact the overall climatic trends as considered by the reviewer.

These important issues have been suitably modified and then added in revised manuscript:

*In addition to data availability, homogeneity also acts as a common concern before using the data. However, such issues should not be considered serious all three types of homogeneity issues namely: volume, instrument type and quality have been addressed before using the data. First, about 5000 radiosonde profiles are available in majority of IGRA stations which are uniformly distributed among all years and seasons (except monsoon); hence it provides a decent data volume for investigation of yearly trends. Secondly, the data of all Indian IGRA stations come from a single type of IM-MK3 radiosondes which has not undergone any change in radiosonde accuracies in the last years and so this addresses the instrument type related issue. Finally, regarding data quality, a set of 7 quality checks are performed by IGRA before accepting the data which should remove any unreliable observations before being used in the study. These 7 quality checks also include repetition check which rejects any possible case of humidity sensor saturation errors during rainy conditions especially in monsoon. Thus it can be inferred that the obtained climatic trends of instability from IGRA is expected to be far more reliable compared to other data sources.*

Ln 96: In 37 years of data, how number of profile is more than~13514 (if there is one profile per day) Lnb 104: Any reason for choosing cubic spline interpolation? Why not linear ?

Reply: The authors accept that they had made a mistake in noting down the dataset volume of radiosonde launches in the table. The total number of radiosonde profiles actually utilized for this analysis is now incorporated in the Table 1A in the revised manuscript. Some discussion about the dataset is also added in the manuscript text.

Cubic spline interpolation has been used because of previous experiences faced by the author during retrieval algorithm testing for temperature and humidity profiling in microwave radiometers. It has been shown in Chakraborty et al. 2016 that linear interpolation cannot reliably

capture the temperature and humidity fluctuations up to boundary layer height. On the other hand piece-wise linear/ quadratic or cubic spline fitting interpolation in retrievals worked comparatively better, hence they were used for further analysis.. This point is now briefly added into the revised manuscript.

*Piece-wise linear/ quadratic/ cubic spline interpolation schemes are employed instead of linear interpolation in temperature and humidity retrievals in this study as the former techniques can more faithfully regenerate the nonlinearities in boundary layer variations of meteorological parameters according to recent studies by Chakraborty et al. (2016)*

Ln105: Calculations of LCL, LFC, EL, CAPE, MLC, CINE etc are the key points of your manuscript. Please mention the formulas in the manuscript. Radiosonde measurement depends on balloon burst altitude. Is the calculation of instability parameters is performed for available radiosonde measurement height or the authors have restricted the iranalysis when the data is available up to minimum height level(like20km). Please discuss those points in detail for each station?

Reply: Since a detailed description of all instability parameters are already given in the supplementary portions, hence the formulae used for calculation of EL, LI, CAPE and CINE are incorporated in that section. LCL, LFC and EL are measured by a parcel theory which requires very exhaustive methodology, hence their corresponding formulae has not been added.

To maintain the reliability and accuracy of instability calculation, radiosonde profiles having balloon burst height above 16 km (~100 hPa) are only considered for the present study for all the Indian IGRA stations. This is now added in the revised manuscript as follows:

*After retrieving the profiles some more additional internal quality checks are performed before using the data for every IGRA station. First, the balloon burst height has to be minimum of 15 km to be selected for analysis.*

Ln 113: Do the radiosonde measure surface wind? How reliable is the surface wind data?

Reply: Radiosondes donot measure surface wind speed. However, the first data available from the balloon ascent is considered to be the surface wind in this study. Normally, these measurements are always within 10m from the actual surface level. These wind speed data is then quality checked and then uploaded to the Wyoming database. According to the correction criterion, wind speeds errors can assume a maximum value of 1m/s up to 100 hPa pressure level. But this 1m/s error is not expected to perturb the thunderstorm severity climate trends in this study as a minimum of 31km/hr or 8.61 m/s surface wind speed is required to be identified as an ordinary thunderstorm.

This has now been added to the revised manuscript as follows:

*Wind speed measurements are taken from the first measurement of radiosonde balloon flight for all stations. These datasets are always within 10m from the surface and according to WMO criterion, they can assume a maximum error of 1 m/s from surface to 100 hPa level. Since a minimum wind speed of 31kmph or 8.61 m/s is required for identification as an ordinary thunderstorm, hence this 1 m/s error is not expected to perturb the presented thunderstorm severity climatology presented in this study.*

Ln131: Why the PCA is performed on the yearly data? Do the analysis is performed for all days or only for TSS and TSO days?

Reply: For the present study, the yearly average datasets are utilized for analysis including PCA. The yearly mean dataset has been used because incorporation of daily datasets would have too many fluctuations which would make the redundant parameter identification very difficult in PCA.

This study has been done for all days of 37 years subject to data availability at 0Z. The daily data is averaged to yearly form for instability climatology while the thunderstorm identification is directly done from the daily near surface wind data.

*To do this analysis in the study, daily datasets of instability parameters are averaged to yearly values for every regions and for all the parameters and then the principal component analysis is performed on the datasets. Daily datasets have not been directly used for PCA as it would have too many fluctuations which would make the redundant parameter identification very difficult in all cases.*

Ln 160: Why the authors need to compare with Chennai? Also couldn't find any comparison dataset? Is this monthly/yearly mean data? What the error bar describe? Why TSO, TSS, WRF and SRF don't have error bar? How trend is calculated? What is the slope value? What information one can extract from such trends? What the positive and negative value means? What is the meaning of increasing CAPE in the atmosphere?

Reply: As already mentioned in the manuscript, the authors want to check the efficacy and reliability of the IGRA long term data by comparing their own highly accurate radiosonde dataset climatology with the nearest IGRA dataset trends; hence Chennai is taken.

For additional comparison, ERA-Interim data has been used as an additional evaluator in this study and both sources support the quality standards in IGRA hence, used in this analysis.

Yearly averaged datasets of instability are utilized here. The error bars show 1 sigma standard error on yearly basis.

The thunderstorm frequency parameters are discrete unit less counts of number of occurrences and not any continuous natural physical parameter. As a result, it was considered safer not to use any deviations using error bars in it.

The trends are calculated using robust fit regression analysis which has already been defined by Andersen (2008).

As already explained the slopes were depicted by robust fit lines to show qualitatively how the instability trends the same data span for nearby stations Gadanki and Chennai match well in spite of using two different datasets. So this supports the use of IGRA datasets for climatological investigations in this study.

Positive and negative values are seen for all parameters as they are normalized with respect to their climatological averages.

Increasing CAPE in the atmosphere signifies more potential energy associated with the updraft leading to bigger convective cells and consequently more severe thunderstorms.

These corrections are added in appropriate sections of the revised manuscript wherever necessary.

*The yearly averaged datasets are normalized with respect to their climatic mean and are plotted along with 1 sigma standard errors in Fig 3 after which robust fit regression analysis (Andersen 2008) is utilized to obtain the climatological trends in these parameters as shown by red solid lines in the plots.*

Ln 168: What height?

Reply: The exact height cannot be ascertained because here a normalized ascent of ~200 hPa is observed whose relationship with height varies for various atmospheric layers.

Ln169: reduction in temperatures near 100 hPa (Fig.3l) plays an important role in modulating the total atmospheric instability and CAPE. Why 100 hPa? Do the authors have any hypothesis to demonstrate it?

Reply: The authors have already explained that the association between 100 hPa temperatures and CAPE had been depicted in many previous attempts over the Indian region like Manohar et al. (1999) and Dhaka et al. (2010).

Ln217: What is ttset analysis?

Reply: Student's ttest is a standard null hypothesis testing algorithm employed to check the statistical significance of the relationship between two datasets one out of which is instability and other is time in years. The basic output of this technique is the p value which indicates the probability of getting outliers in the relationship beyond a certain limit. Values of p less than 0.05 indicate a statistically significant relation. This text is not added in the manuscript due to its marginal application in the entire study. However, a reference has been provided in the manuscript for the readers curiosity.

Ln241: Hence, the mid and upper tropospheric moisture plays a crucial role in modulating the Indian climate. This sentence is not clear.

Reply: This line requires minor corrections because from the previous lines it has been made clear that it is not the lower tropospheric moisture (below 700 hPa) but the remaining amount which is increasing significantly at par with CAPE for all regions, hence there may be a possible association between these two factors which needs to be investigated in the coming sections. This has been added in the revised manuscript.

Ln290: You dataset is for 37 years, how you can get periodicity up to 20 years?

Reply: The authors have already answered to this question previously. In the present study, the time series datasets are not checked by FFT analysis but by EMD and LSP analysis which considers the full time period unlike half of the dataset in the former. Additionally, significant changes in the trends of instability and related parameters are observed most prominently between the years 1996-2004; consequently the relative contribution of this periodicity over the instability trends are found to much higher compared to the other periods. Hence this periodicity has been selected for analysis.

Ln319: trends before and after the period 1996-2000 are significantly different from each other. Any reason? Any change in radiosonde sensor?

Reply: All reviewers have doubt regarding change in sensors around 1998. But this comment is not right because of following reasons.

First, there has not been any mention in past researches about any change in radiosonde data quality during 1996-2004 either from IMD or IGRA. There were some data quality issues in IGRA but they were before the year 1980. Secondly, we have again checked the yearly variations of all 16 parameters for 2 random stations from each sub-division. However, we did not find any abrupt change in time series during the years 1996-2004 except a few cases. This argument can also be validated from the climate trends in instability from Chennai in Figure 3. Third and most importantly, reports from IMD have clearly mentioned that the year 2000 was a

"tipping point" for the impact of climate change-led warming in the country and situations are going to be the worst by 2040 if appropriate steps are not taken to address the various sources of emissions which are the main cause for this intense global warming (https://www.hindustantimes.com /environment/freak-weather-to-rise-in-india-over-two-decades/storyT1G8SgfBh8jydT15UnKGuM.html). Hence this explains how the climate trends have changed drastically since the end of the last century and this is not due to change in the radiosonde sensor.

These arguments are also briefly added in conclusion section of the manuscript to clear further doubts among the readers in future which is as follows:

*After going through the study, there may be a possibility of thinking that the change in instability trends is due to the change in sensors around 1998. But this is not the actual case because first, there has not been any mention in past literature survey related to any change in radiosonde data quality during late 1990s in IMD or IGRA. Secondly, the yearly variations of all 16 parameters for various IGRA stations as in Chennai do not commonly show any abrupt change in time series during 1996-2004 except for a few cases. Thirdly it has been revealed by IMD reports that the year 2000 was a tipping point for the climate change led warming over India thereby leading to a rise in catastrophic weather events and cataclysmic fallout will follow by the year 2040 if these emission scenarios are not curbed recently (Hindustan Times, 2019). Thus, it follows that the observed changes in the atmospheric instability trends before and after 1996-2004 are due to a synoptic global warming based climate change phenomena and not due to any change in radiosonde sensor type.*

We thank the reviewer once again for providing detailed his suggestions which helped us improve the manuscript content further.
* * *
**Replies to Reviewer #3 Comments/Suggestions**

First of all we wish to thank the reviewer for providing constructive comments/suggestions which significantly improved the content of the manuscript. The authors have addressed all the comments raised by the three reviewers and incorporated in the revised manuscript.

Main Comments:

Figure 3d: The changes in EL, LI, and CAPE between 1980 and 2016 are difficult to believe and if true compelling. Have other studies shown such huge changes?

Reply: There have been multiple papers which reported highly increasing trends of various thermodynamic instability parameters both globally and over India in the last few years. Gettlemann et al (2002) has showed a ~20% increase per decade in CAPE from tropical radiosonde stations globally during 1958-1997. Murugavel et al. (2012) has shown a steep rise in CAPE during the monsoon season of 1984-2008 with a slope of 38 J/kg over India. Thus, a very prominent rise in CAPE over India in the last 37 years is not an unexpected result. Some portions of this have been incorporated in the revised manuscript as follows:

*It may appear at certain sections of this analysis that the trends of CAPE and EL are exorbitantly high; but it is not the actual case because previous studies by Murugavel et al*

*(2012) and Gettlemann et al. (2002) have also shown almost comparable trends in convective severity both in India and abroad.*

Changes in aerosol loading and subsequent changes in the morphology of clouds due to the aerosol indirect effect are not discussed and should be considered when examining trends in stability and precipitation over the Indian region. Please discuss the role of aerosol forcing may play in explaining these trends.

Reply: In this study, we have tried to understand the association of several natural and anthropogenic factors responsible for the drastic growth in atmospheric instability and thunderstorms in the recent years. In this connection, there also has been an attempt to show the direct forcing effect of aerosol components in generating a weak inhibition to convective activities as already discussed by previous researches. But on the other hand, the relationship between indirect aerosol forcing and instability is comparatively complex (Connoly et al. 2012). Some recent studies have revealed that a higher concentration of aerosols may lead to stronger updrafts velocities by altering the latent heat release resulting in growth of CAPE and TSS (Tao et al. 2012; Storer and van den Heever, 2013). However, this is a season and location specific phenomena and hence it is not expected to impact the yearly trend of CAPE and TSS as strong as the upper tropospheric cooling effect projected in this study. But in future, an exhaustive analysis of cloud and aerosol components involving both in-situ and modelled data can to be done to investigate its contribution on the total CAPE, TSS and SRF trends over the Indian region. This has been mentioned in the conclusion section of the revised manuscript as follows:

*On the other hand, this study also introduces the effect of direct aerosol heating on instability and convection; but the probable impact of indirect aerosol loading in modulating the cloud lifetime and convective severity has not been discussed here. This is because, the relationship between indirect aerosol forcing and instability is still unclear and complex (Connoly et al. 2012). A few researches in the recent years have hypothesized that a higher concentration of aerosols may lead to stronger updrafts velocities by altering the latent heat release resulting in growth of CAPE and TSS (Tao et al. 2012; Storer and van den Heever, 2013). However, this is a season and location specific phenomena and hence it is not expected to impact the yearly trend of CAPE and TSS as strong as the upper tropospheric cooling effect projected in this study. But in future, an exhaustive analysis of cloud and aerosol components involving both in-situ and modelled data can to be done to investigate its contribution on the total CAPE, TSS and SRF trends over the Indian region.*

The authors look at trends in 16 different variables derived from radiosonde data. That makes for a difficult read. Might make sense to condense the variables to 8-10 by removing highly correlated variables.

Reply: We have considered reducing the number of parameters shown in the manuscript but ended up with the dilemma that for a complete understanding about the morphology of upper tropospheric instability, all the instability parameters are required; hence the number of parameters used in this study cannot be condensed. However, to reduce the confusion of readers, main parameters like LFC, EL, CAPE, CINE, PWV, T100, TSS and SRF are to be retained in

the main figures while their complementary aspects such as LCL, LI, VT, MLC, PWL, WSH, TSO and WRF are kept in supplementary sections. This has now been mentioned in the revised manuscript as follows:

*However, it should be noted that Fig 3 provides too much detailed and cumbersome results related to all 16 parameters and the complexity of the analysis is expected to increase further when similar analysis will be presented for all the Indian regions together.  But on the other hand, for a complete understanding about the morphology of upper and lower tropospheric instability, all the instability parameters will be required. Hence, to reduce further chances of confusion and to make the results more compact, all 16 parameters will be discussed together but only a few of them will be presented in the main study. After a thorough consideration with respect to the main objective of the present attempt, 8 parameters namely LFC, EL, CAPE, CINE, PWV, T100, TSS and SRF are to be retained in the main figures while their complementary aspects such as LCL, LI, VT, MLC, PWL, WSH, TSO and WRF are shown in the supplementary sections.*

It is difficult to determine the regional trends from the plots. Perhaps the means/trends for the 3 regions (coastal, interior, and other) can be separated vertically as opposed to stacked on top of each other in plots 4-6 and 8-9.

Reply: The authors humbly suggest that showing the means/trends of these three common sub-division types is not reasonable as it will become difficult to compare the difference between these regions.  On contrary, the existing approach makes it easier to identify that coastal regions have more variability and trends compared to the others. Further, with the reduction in number of subplots, the readability of the figures has also improved significantly; hence now it may not be difficult to understand the regional trends.

Specific Comments L36 to L48: I would suggest limiting your references to studies that focused on India. Alternatively, you need to explicitly state for what region and what time period the results you cite are valid.

Reply: The authors have tried to give equal preference to studies both within and outside India to highlight the global perspective of the research problem. As per reviewer suggestion the references which are very much similar to our studies are only added in the revised version.

*Over the Indian region, Manohar et al. (1999) studied the latitudinal variation and distribution of thunderstorm frequency and CAPE over 78 Indian stations during 1970-1980 and he postulated that the ambient temperature at 100 hPa pressure level has a strong relationship with it. Dhaka et al. (2010) utilized radiosonde observations during 1958-1997 and obtained very prominent autocorrelations on both yearly and seasonal basis between convection strength (CAPE) and upper troposphere temperatures at 100 hPa (T100). Later, Murugavel et al. (2012) studied the long term trends of CAPE from 32 radiosonde stations during 1984-2008 and revealed an alarming growth in monsoon CAPE over India with a slope of 38J/kg/year. However, they additionally stated that the low-level moisture and solar cycle can have additional impact on the increasing CAPE. Recently, using reanalysis datasets Chakraborty et al. (2017a) and Saha et al. (2017) reported that lower tropospheric instability is reducing over few Indian stations after 1980 due to increasing levels of pollution.*

L54: Increases in air pollution and greenhouse warming may have opposing effects on lower atmospheric stability. Don't group them together here.

Reply: The authors have made the necessary corrections in the revised manuscript as:

*Recently, Chakraborty et al. (2017a) and Saha et al. (2017) reported that lower tropospheric instability is reducing at certain Indian stations predominantly due to pollution.*

L88-90. This sentence is confusing. Are you saying that typically a station has 2-7 gaps with each gap being less than one month in length? If yes, please say so.

Reply: The authors have clarified the confusion created by these lines in the revised manuscript as follow:

*When an in depth investigation is done on the data continuity by plotting the temperature and humidity profiles for all days, a set of monthly gaps were noticed in the data. Most of the stations had data gaps of 2-7 days in some months but, on the whole, except for very few cases, the duration of these individual data gaps are mostly limited to less than 1 month. However, these small data gaps are not expected to provide any significant impact on the long-term seasonal or annual average variation of (37 years x 12 months) span.*

Figure 1: Identify the sites with the serial number from Table 1A.

Reply: Necessary corrections have been done in the revised manuscript.

Figure 2: Sufficient space is available at the top of each plot to replace the acronyms with the actual names, e.g., CI –> Central India

Reply: The authors have modified the figure caption text to remove any possible confusion.

Figure 4: This set of plots confuses me. a) By inspecting these plots, is it possible to separate the west coast from the east coast and central India from Peninsula-India? b) Shouldn't there be a separate box and whisker plot for each region? c) I'd suggest flipping the vertical pressure coordinates so that high pressures are located near the bottom and low pressures near the top. d) How can the mean for a region be located at the 5th or 95th percentile? e) Why do I only sometimes see "whiskers"?

Reply: (a) The number of subplots has been reduced to half now hence the symbols corresponding to various regions can be easily being resolved and understood. (b) Grouping and separation of regions cannot be done as already explained before. (c-d) This has been done in the revised figures. (d) The average values of parameters for all six sub-divisions along with total Indian region average are shown in form of a distribution to give a feel to the readers about the general instability conditions across the Indian region. The regions in the percentiles indicate more extreme weather conditions on average. (e) If a quartile is present very close to the total mean value, then no whisker can be seen, hence these visual effects.

L114-116: Is rainfall or lightning required or is the determination strictly based on wind speed?

Reply: IMD reports strictly classify thunderstorm intensities based on the basis of maximum surface wind speeds (http://imd.gov.in/section/nhac/termglossary.pdf).

L135-136: "VT is found to lie exactly in the middle ..." Arguably, TTI or CT is more in the middle than VT.

Reply: It is true that VT is not solely at the centre in all cases. However, as VT indicates the temperature difference between 850 and 500 hPa hence, it better represents the lower

tropospheric instability compared to other parameters. Consequently, VT has been given additional preference for selection towards the long-term analysis. This has been mentioned in the revised manuscript as follows:

*Since VT is found to lie exactly in the middle of the rest of the parameters, and it also represents the lower tropospheric instability in a much more suitable way; hence this parameter can also be used to represent the population in a convenient way.*

L163: I do not see a trend in PWL. C2

Reply: The discussion on PWL trend has been removed from that line as per reviewer comments.

*A decreasing trend in VT and a strengthening of CINE with LI is noticed which indicates a reduction in the lower atmospheric instability.*

L168: "intensification in EL". This is confusing. Go with "increase in the height of the EL".

Reply: This has been corrected in the revised manuscript.

Table 1: Since the p tests always yield the same results, i.e., significant , I would suggest replacing those columns with columns that indicate the percent trend. I would also suggest adding a column that indicates the units.

Reply: This has been corrected in the revised manuscript and the corrected lines are as:

*The p values are calculated at 95% confidence limits for ttest analysis on all instability parameters over the Indian sub-divisions and interestingly, all the values are found to be below 0.05. Hence the time series variations to be presented in subsequent sections will always be statistically significant in nature. So, to have a better quantitative measure of the trend significance, the total changes in each of these parameters are presented in percentage form in place of the p values in the table. This process will enable an easy identification of regions experiencing more accelerated convective growth. But on the other hand, while analyzing the results of the trend analysis in statistical form, the absolute trend has to be given more importance as the % change depend on the magnitude of the long term mean.*

L247: "TSS is found to increase drastically". What are the units for TSS. What do you mean by increasing drastically. L261-262: Chicken and egg question: Is "more convective rain" the cause or consequence of changes in the LFC?

Reply: TSS is the frequency of severe thunderstorms and it does not have any unit. By "drastic" it meant to depict that the change in TSS is much higher compared to TSO.

*However, the TSS (Fig.5g) is found to increase at a much higher rate compared to TSO especially in the coastal regions.*

This line has been modified to get rid of extra confusion.

*However, this ascent is more prominent in the monsoon and post-monsoon season.*

L271: What does it mean for T100 to strengthen?

Reply: The authors wanted to refer to a "small cooling effect" there. The word "strengthen" has been replaced in the revised manuscript.

L290: I find it questionable to look for periodicities of 16-20 years in a data set that is only twice that long.

Reply: In this study, the time series datasets are not checked by FFT analysis but by EMD and LSP which considers the full time period unlike half of the dataset in the former. Additionally, significant changes in the trends of instability and related parameters are observed most prominently between the years 1996-2004; consequently the relative contribution of this periodicity over the instability trends are found to much higher compared to the other periods. Hence this periodicity has been selected for analysis.

L318: "Drastic" can mean different things to different people perhaps use a different adjective. Also, be specific as to which instability parameters showed "drastic" changes.

Reply: This line has been modified in the revised manuscript as follows:

*In the previous section, the annual averaged time series of many parameters such as EL, LI, VT, CAPE, CINE, T100, TSS, WRF and SRF has showed very significant changes with respect to MCO.*

L325: EL –> EL height

Reply: This line has been modified in the revised manuscript.

L356: ozone breakup –> ozone decreases

Reply: This line has been modified in the revised manuscript.

L358: cooling effect –> cooling effect due to a reduction in downwelling long wave radiation

Reply: This line is correct and it does not need any correction.

L395: What do you mean by "strong cooling due to ozone decomposition?"

Reply: This line has been corrected in the revised manuscript as follows:

*…the upper troposphere and lower stratosphere (UTLS) where it undergoes prominent cooling due to ozone reduction.*

L398: Why would the dearth of transported moisture affect that rate of pollutant dispersion by the winds?

Reply: This line was wrongly written as moisture has nothing to do there. Changed as:

*However, in the inland regions the layer of absorptive aerosols and greenhouse gases cannot be dispersed amply due to the dearth of strong lower level winds.*

L423: CAPE increases in all regions not just near the coast. Please rephrase bullet point 1. Also, "suffer" is a poor choice of words.

Reply: These lines have now been corrected in the revised manuscript as follows:

*The coastal regions experience the most significant increase in Convective Available Potential Energy.…*

L426: "drastic" is a qualitative term - be more quantitative C3

Reply: This word has replaced as "significant" in the revised manuscript.

L439: Are you certain this leads to a strong cooling effect in the troposphere? The increases in OH would lead to increases in the oxidation rates of CO and methane, which could lead to more ozone in the presence of NOx.

Reply The authors humbly accept that they are no experts in UTLS chemistry. However, a cooling in UTLS, reduction of ozone and a simultaneous increase in water vapor indicates some complex mechanism initiated by moisture intrusion which eventually leads to ozone breakup. On the other hand, as pointed out by the reviewers, OH molecules may also have a probability to both increase and decrease the ozone concentration. So, to clarify this confusion, the phrase involving OH radical effect has now been faded out of the conclusion section. The modified conclusion result is as follows:

*In the coastal regions, ample amount of water vapor is advected into the mid-troposphere from the surrounding seas which in presence of strong lifting goes up to upper troposphere and lower stratosphere (UTLS) where ozone depletion occurs leading to a strong cooling effect. This cooling effect enables the ascent in EL resulting in much stronger LI and CAPE values, hence more TSS and SRF.*

TEXT S1: S6: Free condensation –> Free Convection S11: Parcel may continue moving past the EL due to upward momentum. S24: Add a reference to the supercell comment S25-32: "lifted from the LFC to the lowest 100 mb of the troposphere". This is incorrect. Please re-phrase this. I believe the moisture and temperature profiles are averaged over the lowest 100 hPa and then the resulting parcel is lifted to the LCL. S42: calculated as th –> calculated as the S48: resaerch –> research

Reply: All these corrections have been done in the revised manuscript.

Minor comments

L32 showed –> shown L39: due to surface heating –> due to increases in surface heating L46: extreme precipitations –> extreme precipitation events L46: intense convections –> intense convection L54: lower instability is reducing –> lower tropospheric instability is decreasing L58: studies over India has –> studies over India have L108: upto –> up to L171: reduction VT –> reduction in VT L192: higher in the coasts –> higher at the coasts L195: "higher", Do you mean "more negative?" L217: "ttset" – > test L223: "all the regions" –> not true in the NE region L224: is minimum –> is smallest L227: "also show an enhancement " –> become more negative L243-244: Smallest changes in the NE and NW regions. The difference between inland and coast regions isn't that large (2 versus 2.375 degrees) L376: a dominant increase –> an increase L380: resulting more –> resulting in more L382: To prove this hypothesis –> To test this hypothesis L383: increasing prominently –> increasing L383: expand DLWRF acronym

Reply: All these corrections have been done in the revised manuscript.

We thank the reviewer once again for providing detailed suggestions which made us to improve the manuscript content significantly.

---

## Author Response (AR2)

**Replies to Reviewer #1 Comments/Suggestions**

First, we wish to thank the reviewer for providing comments/suggestions which improved the content of the manuscript further. The authors have addressed all the comments raised by the reviewer and incorporated in the revised manuscript.

Main Comments:

Page 2, line 59-60: Improve sentence: "Saha et al. (2017) reported that lower lower tropospheric instability is reducing".

Reply: This line has been modified in the revised manuscript as follows:

*Chakraborty et al. (2017a) and Saha et al. (2017) reported a weakening in lower tropospheric instability over few Indian stations due to increasing pollution levels using reanalysis datasets.*

Page 4, line 130: Cut one dot "(2016)..".

Reply: This correction has been done in the revised manuscript.

Page 4, line 149: Add a blank between 31 and kmph ("31kmph").

Reply: This correction has been done in the revised manuscript.

Page 5: line 177: Cut one dot "way..".

Reply: This correction has been done in the revised manuscript.

Page 7, line 247: Change "from Sea" to "from sea".

Reply: This correction has been done in the revised manuscript.

Page 7, line 272: Change "a t-tset analysis" to "a t-test analysis"

Reply: This correction has been done in the revised manuscript.

Page 8, line 302: Improve sentence "significantly at par with CAPE"

Reply: This line has been modified in the revised manuscript as follows:

*Hence it follows that it is not the lower tropospheric moisture (below 700 hPa) but the remaining amount which is increasing significantly for all regions. Now, as this growth in upper tropospheric moisture is analogous with a parallel rise in upper level CAPE, hence there should be a possible association between these two factors which needs to be investigated.*

Page 20, Fig. 1: Change "4 stations in the NC" to "4 stations in the CI"

Reply: This correction has been done in the revised manuscript.

We thank the reviewer once again for the help in pushing us forward to improve the manuscript content.